# Influence of Carbon Source and Iron Oxide Minerals on Methane Production and Magnetic Mineral Formation in Salt Marsh Sediments

Kaleigh R. Block[1], Amy Arbetman[2], Sarah P. Slotznick[3], Thomas E. Hanson[1], George W. Luther III[1], Sunita R. Shah Walter[1]

[1]School of Marine Science and Policy, University of Delaware, Lewes, DE, USA
[2]Macalester College, Saint Paul, MN, USA
[3]Department of Earth Sciences, Dartmouth College, Hanover, NH, USA

*Correspondence to*: Sunita R. Shah Walter (suni@udel.edu)

**Abstract.** Salt marshes can emit significant methane to the atmosphere. These emissions are highly variable, but the cause of this variability is not well-understood. Although methanogenesis should be inhibited by sulfate reduction where sulfate is present due to its thermodynamic unfavorability, methane emissions are not well-predicted by sulfate concentrations and other controls on methane production must be active. One hypothesis is that where sulfate is present, salt marsh methanogens are fueled by methylated carbon substrates that sulfate reducers do not compete for. It has also been shown that crystalline iron minerals can facilitate increased methane production in low-salinity wetlands, but this has not been explored in salt marshes. This study documents how different organic carbon sources (monomethylamine and ethanol) and Fe(III) minerals (ferrihydrite, magnetite and hematite) influence methane production by microbial communities from a polyhaline tidal marsh creek in the Great Marsh Preserve, DE, USA. Carbon source had a strong influence on microbial community composition by the end of the incubations. More methane was produced with monomethylamine amendment than with ethanol, and the highest methane production rates were in incubations supplied with both monomethylamine and magnetite or hematite. This increased methane production in the presence of (semi)conductive iron minerals could indicate that interspecies electron transfer was active in some of our treatments. But instead of the more commonly described syntrophic partners, this interaction appears to be between methylotrophic methanogens belonging to *Methanococcoides* and an unidentified iron reducing bacterial group, possibly *Ca*. Omnitrophus. Much less methane was measured with ethanol and ferrihydrite amendments. But in ethanol-amended incubations, a small proportion of anaerobic methane oxidizers was detected, which suggests both methane production and re-oxidation may have occurred leading to low measured methane production. Although some iron reduction and $Fe^{2+}$ production was observed in all treatments, significant transformation of ferrihydrite to magnetite was observed only with ethanol amendment. If microbially-mediated magnetite formation occurs in salt marsh sediment, our observations indicate that the resulting magnetite could enhance methane production by methylotrophic methanogens. This study highlights the importance

of methylated compounds to salt marsh methane production as well as the potential importance of iron mineral composition for predicting methane production and iron reduction rates.

## 1 Introduction

Wetlands are the primary natural source of methane emissions to the atmosphere (Saunois et al., 2016; Zhang et al., 2017). However, only a fraction of these emissions is expected to come from high-salinity tidal wetlands such as salt marshes due to

the influx of oxygen and sulfate during regular inundation (Bartlett et al., 1987; Hartman et al., 2024; Holmquist et al., 2023; Poffenbarger et al., 2011). According to the textbook paradigm, this is because thermodynamic considerations determine the dominance of microbial metabolisms. Organic carbon degraders are expected to follow the redox ladder, sequentially depleting oxygen, nitrate, manganese and iron oxide minerals, and sulfate before methanogenesis is favored (Jørgensen, 2006; Jørgensen and Kasten, 2006; Sundby, 2006). In salt marshes, oxygen is often depleted within millimeters of the sediment surface and

sulfate reduction is responsible for a large fraction of organic carbon remineralization (Howarth, 1984; Howarth and Giblin, 1983; Howarth and Teal, 1979), as well as the oxidation of methane (Krause and Treude, 2021; Segarra et al., 2013). However, in more oxygenated marsh regions like creek banks and bioturbated zones where pore waters are more efficiently flushed by the tides, iron reduction can be the dominant microbial metabolism (Antler et al., 2019; Gribsholt et al., 2003; Kostka et al., 2002a, b; Krause et al., 2025). Iron oxides may also fuel anaerobic methane oxidation in salt marshes (Krause and Treude,

2021; Liu et al., 2025; Segarra et al., 2013). Salt marsh sediments can also have high iron concentrations with seasonally-fluctuating mineralogy (Kostka et al., 2002a, b; Kostka and Luther, 1994, 1995; Taillefert et al., 2007). Based on energy yield, the presence of measurable iron oxide minerals and sulfate should inhibit methanogenesis as well as facilitate the anaerobic oxidation of methane produced at deeper depths before it is released to marsh creeks and the atmosphere.

At the ecosystem scale, methane emissions data from wetlands do follow a general decreasing trend with increasing sulfate

concentrations (Al-Haj and Fulweiler, 2020; Chmura et al., 2011; Hartman et al., 2024; Poffenbarger et al., 2011), but significant variability in methane emissions across salinity gradients has also been reported, indicating the importance of other controls (Al-Haj and Fulweiler, 2020; Bridgham et al., 2006; Chmura et al., 2011; Hartman et al., 2024; Rosentreter et al., 2021). In the mid-Atlantic region of the U.S., high concentrations of methane and indications of active methanogenesis were found to coexist with sulfate reduction in marsh sediments (Seyfferth et al., 2020). This finding indicates thermodynamics

does not entirely predict the activity of microbial metabolisms and biogeochemical processes in salt marshes. Methanogens have been shown to co-exist with sulfate reducers, although with lower activity (Sela-Adler et al., 2017; Senior et al., 1982). They have also been shown to circumvent the redox ladder through utilization of organic substrates that are not as available to sulfate-reducing bacteria such as methylated compounds (Buckley et al., 2008; Oremland et al., 1982; Parkes et al., 2012a; Winfrey and Ward, 1983; Yuan et al., 2019). In the eastern U.S., the low marsh is dominated by the native marsh grass,

*Spartina alterniflora,* which releases trimethylamine at depth through root exudates and root degradation (Wang and Lee, 1994a). Trimethylamine, dimethylamine and monomethylamine have all been detected in the pore waters and sediments of

salt marshes (Fitzsimons et al., 1997; Parkes et al., 2012b; Summons et al., 1998; Wang and Lee, 1994b). As sulfate-reducing bacteria are not known to utilize methylamines (Mausz and Chen, 2019), this supply of carbon substrate from *S. alterniflora* allows methanogenesis to proceed simultaneously with sulfate reduction (Seyfferth et al., 2020; Yuan et al., 2014, 2019).

Methanogens can also skip rungs of the redox ladder by participation in syntrophic relationships with iron-reducing bacteria (Cruz Viggi et al., 2014; Kato et al., 2012; Rotaru et al., 2014, 2021; Xu et al., 2019; Zhuang et al., 2019). Although this has not yet been observed in salt marsh microbial communities, methanogens can obtain their reducing equivalents through interspecies electron transfer (IET) facilitated by conductive iron minerals (Lovley, 2017a, b; Rotaru et al., 2021). Although the majority of iron in shallow marsh sediments undergoes a seasonal alternation between amorphous Fe(III) minerals and iron

sulfides at shallow depths (Koretsky et al., 2003; Kostka and Luther, 1994), crystalline Fe(III) minerals with (semi)conductive properties that could facilitate IET have been shown to persist year-round, even at anoxic and sulfidic depths (Kostka and Luther, 1994, 1995; Taillefert et al., 2007). It has also been shown that ferrihydrite, an amorphous Fe(III) mineral, can be converted to the secondary conductive mineral magnetite over time in anoxic incubations of microbial communities from rice paddy soils, eventually enhancing methane production through IET (Tang et al., 2016; Zhuang et al., 2015, 2019). Although

they have not been as extensively studied as sulfate reducers, iron-reducing bacteria have been reported in salt marsh sediments (Hyun et al., 2007; Koretsky et al., 2003, 2005; Lowe et al., 2000). These populations have the potential to cooperate with methanogens through IET or to outcompete them and transform Fe(III) minerals into aqueous $Fe^{2+}$, mixed-valent minerals like magnetite, or Fe(II) minerals like pyrite. Salt marshes, with their naturally variable redox conditions, have also been found to host greigite-producing magnetotactic bacteria, often in sediments at the transition between oxic and anoxic conditions

(Bazylinski et al., 1991; Blakemore, 1975; Demitrack, 1985; Simmons and Edwards, 2007a, b; Sparks et al., 1989). And both magnetite- and greigite-producing magnetotactic bacteria have been found in salt ponds neighboring salt marshes (Simmons et al., 2004; Simmons and Edwards, 2007b).

In this study, we investigate how specific carbon sources and Fe(III) minerals influence the production of methane and magnetic minerals in anoxic, sulfate-free incubations of marsh creek sediments from a polyhaline salt marsh in the mid-Atlantic

region of the eastern U.S. We studied non-vegetated marsh creek sediments adjacent to a marsh platform dominated by *S. alterniflora* to avoid the multiple and confounding influences of marsh grass stems and roots on redox conditions and methane emissions (Määttä and Malhotra, 2024). We investigated the effect of adding organic carbon substrates for methylotrophic and acetoclastic methanogenesis combined with iron minerals that supply Fe(III) with a range of reactivity and electrical conductivity. Here we compare methane production from monomethylamine (MMA), which represents methylated

methanogenic substrates (Krause and Treude, 2021), to methane production from ethanol (EtOH), which is oxidized to acetate through fermentative pathways (Schink, 1997) and which has been used in many studies investigating the potential for magnetite and hematite to promote electric syntrophy between methanogens and bacteria in soil and sediment enrichments (Kato et al., 2012; Rotaru et al., 2019, 2018). MMA and EtOH amendments were combined with additions of magnetite, hematite, and ferrihydrite to determine the influence of these iron minerals on methane production and iron reduction by salt

marsh microbial communities.

## 2 Materials and Methods

### 2.1 Study Site Description and Sediment Sampling

Sediments for this study were sampled from the Great Marsh Preserve in Lewes, DE, USA. In this location, low marsh is dominated by the marsh grass *S. alterniflora* and it is dissected by a grid of marsh creeks originally dug as mosquito ditches in the 1930's but left to return to natural conditions by the 1970's. The marsh creeks and low marsh platform regularly experience inundation at high tide with water from the lower Delaware Bay via the Roosevelt Inlet and Canary Creek. High tide salinity averages 30 PSU during the month of June (2016–2020 average; Delaware Water Quality Portal Station BR40, 2024), and the tidal amplitude averages 1.2 meters (Dewitt and Daiber, 1973; Guider et al., 2024). Sediments for our study were collected by push core from a mud flat adjacent to the low marsh platform within a marsh creek (38.78926° N, 75.16968° W) while it was covered by shallow water at low tide. The mud flat was selected to avoid living *S. alterniflora* stems and roots. Two sediment cores of 8.5 cm diameter and 22.5 cm length were taken on June 29, 2023. They were processed for geochemistry within five hours of collection.

### 2.2 Downcore Profiling of Redox Conditions

Before the sediment core was extruded in the laboratory, cyclic voltammetry (CV) was used to determine concentrations of dissolved $O_2$, $\Sigma H_2S$, $\Sigma S^{2-}$, $Fe^{2+}$, $Fe^{3+}$, and $FeS_{aq}$ molecular clusters on the intact core as well as qualitative concentrations of solid Fe(III) nanoparticles and organic-Fe(III) aggregates. CV measurements were made with a three-electrode system composed of a solid-state gold amalgam (Au/Hg) working microelectrode encased in glass, a silver-silver chloride (Ag/AgCl) reference electrode and 0.5 mm platinum wire counter electrode (Brendel and Luther, 1995). Electrodes were coupled to a model DLK-60 electrochemical analyzer (Analytical Instrument Systems Inc, Flemington, NJ, USA). The reference and counter electrodes were secured to the core barrel such that they were suspended in marsh creek water, approximately 2 cm of which was retained above the sediment in the core barrel. The working electrode was lowered into the core at fine vertical resolution by a micromanipulator (Luther et al., 2008). Concentrations of $\Sigma S^{2-}$ and $O_2$ were calculated from current using published parameters (Luther et al., 2008; Waite et al., 2006). Between the sediment-water interface and 0.35 cm depth, measurements were taken every 0.05 cm. Below 0.35 cm, measurements were taken every 0.1 cm until a depth of 1.2 cm. Since the redox transitions were detected at shallower depths, CV measurements were discontinued at a depth of 1.2 cm. All scans were made in triplicate.

### 2.3 Ferrihydrite Synthesis

Ferrihydrite was synthesized following the methods of (Smith et al., 2012). Ferric nitrate nonahydrate (20.25 g) and ammonium bicarbonate (11.91 g) were ground together with a mortar and pestle. The product of this reaction was dried overnight, rinsed

the following day via vacuum filtration with small aliquots of deionized water and dried again overnight. The final product was ground again with a mortar and pestle and used for ferrihydrite amendments in incubations.

## 2.4 Incubations

Although we observed a redox transition at ~0.5 cm, sediments were unconsolidated at this depth. To avoid mixing soluble oxidants (e.g., oxygen, nitrate, sulfate) from shallower depths into sediments used for incubations from below the redox transition, the top 2.5 cm of the core was discarded following CV measurements (Fig 1a). Sediments between 2.5 cm to 16.5 cm depth provided sufficient sediment volume for our incubations and this interval of sediment was extruded and homogenized in a 2.7 mil thickness, low-density polyethylene (LDPE) zip-top bag. The small amount of dead plant material within the core was removed by hand picking during extrusion and transfer to the bag. Homogenized sediments were refrigerated at 4° C for ~16 hours prior to mixing sediments into a slurry and assembling the incubation bottles. The slurry was made up of 1:3 v/v homogenized marsh sediment and artificial sulfate-free seawater. Sulfate was not included in our artificial seawater to disadvantage sulfate reduction in favor of other microbial metabolisms and allow investigation of controls on methane production other than competition with sulfate-reducing bacteria. Artificial sulfate-free seawater contained 26.4 g NaCl, 11.2 g MgCl$_2$·6H$_2$O, 1.5 g CaCl$_2$·2H$_2$O and 0.7 g KCl per liter (Aromokeye et al., 2018) as well as 1 mM of a an iron-free trace metal solution modified from DSMZ Trace Element Solution SL-10 (Koblitz et al., 2023): 70.0 mg ZnCl$_2$, 100 mg MnCl$_2$·4H$_2$O, 6 mg H$_3$BO$_3$, 190 mg CoCl$_2$·6H$_2$O, 2 mg CuCl$_2$·2H$_2$O, 24 mg NiCl$_2$·6H$_2$O, and 36 mg Na$_2$MoO$_4$·2H$_2$O per liter. Anaerobic incubations were performed in 125 mL serum bottles containing 50 mL of slurry and 75 mL of headspace. The headspace was purged of O$_2$ by flushing for 2 minutes with N$_2$/CO$_2$ (80/20) prior to sealing the bottles with an N$_2$/CO$_2$ (80/20) headspace. Incubations were supplemented with 20 mM organic carbon, either ethanol (EtOH) or monomethylamine hydrochloride (MMA). Incubations were also supplemented with 20 mM powdered iron minerals, either synthesized ferrihydrite, magnetite, or hematite, or no iron. Powdered magnetite was purchased from Sigma-Aldrich (St. Louis, MO, USA), and powdered hematite was purchased from Strem Chemicals Inc. (Newburyport, MA, USA). In total, 24 bottles were prepared with the following treatment groups in triplicate: EtOH + no iron, EtOH + ferrihydrite, EtOH + magnetite, EtOH + hematite, MMA + no iron, MMA + ferrihydrite, MMA + magnetite and MMA + hematite. Incubations were maintained at 30° C under static conditions in the dark for 34 days.

## 2.5 Methane and Ferrous Iron Measurements

Headspace methane concentrations were measured twice weekly by gas chromatography with a flame-ionization detector (Agilent 7890B, Wilmington, DE, USA). Methane was separated isothermally on a HP-Plot Q column (Agilent, 30 m x 0.530 µm inner diameter, 40 µm film thickness). Headspace samples were manually injected with a gastight syringe in split mode with split ratio of 5:1. The injector temperature was 250° C, oven temperature was 60° C and detector temperature was 250° C. Methane peaks were integrated using Agilent ChemStation Data Analysis software version F.01.03. A parallel set of 125 mL serum bottles were prepared as methane standards with the same ratio of distilled water to headspace as incubation bottles to

account for dissolved methane. The headspace of the standard bottles was filled with 80/20 $N_2/CO_2$ and varying amounts of methane. Six standards with total methane concentrations ranging from 0.004 to 0.4 mmol were injected with each set of headspace methane measurements to obtain calibration curves. Methane concentrations and calculated methane production rates in incubations (Figs. 2, 3) were determined from a calibration curve of equilibrated standards and the ideal gas law. Measurements of dissolved ferrous iron were made twice weekly. Ferrous iron was quantified utilizing the ferrozine assay method developed by (Stookey, 1970). From each incubation, 0.5 mL of slurry was sampled from the bottles, filtered at 0.45 µm and stabilized in ferrozine solution. Absorbance at 562 nm was measured with a HP-8452A Diode Array spectrophotometer (Hewlett-Packard/Olis Inc., Athens, GA, USA).

## 2.6 DNA extraction, PCR, quantification, and sequencing

At the end of the incubation period, slurries were homogenized and subsamples were taken for DNA extraction. One bottle was selected at random from each treatment group for amplicon sequencing. DNA was extracted from the slurry using the DNeasy PowerSoil Pro Kit (QIAGEN, Germantown, MD, USA). Concentrations of DNA were quantified with a Qubit BR DNA assay (Invitrogen, Eugene, OR, USA) using a DeNovix DS-11 fluorometer (Wilmington, DE, USA). For all extractions, the total amount of DNA recovered was ≥ 6 ng. Amplification of extracted DNA was performed with a modified two-step barcoding method (Herbold et al., 2015; Tadley et al., 2023). Amplicons of the V4 region of the 16S rRNA gene were produced with the primers 515F (Parada et al., 2016) and 806R (Apprill et al., 2015) modified with the head sequence: 5'-GCTATGCGCGAGCTGC-3' (Herbold et al., 2015). The second PCR barcoding primer was paired to the head sequence. Each PCR reaction contained 1.0 µM forward and reverse primers in Invitrogen SuperFi Master Mix (Thermo Fisher Scientific, Waltham, MA, USA) with denaturation at 95° C for 30 seconds, annealing at 60° C for 30 seconds and extension at 72° C for 60 seconds for a total of 30 cycles in the first PCR and 5 cycles for the second PCR. During the first PCR, reactions were completed in triplicate and PCR products were screened by gel electrophoresis. The barcoded amplicons were purified with sparQ PureMag beads (QuantaBio, Beverly, MA, USA). Sequencing on an Illumina MiSeq platform (MiSeq Reagent Kit v2, 2 x 250 bp, San Diego, CA, USA) was performed at the University of Delaware Sequencing and Genotyping Center.

Amplicon sequences were analyzed as described by (Tadley et al., 2023). Sequences were sequentially processed with Python (3.10.10) and QIIME2 (2021.2; Bolyen et al., 2019) for demultiplexing as described by Herbold et al. (2015). Amplicon sequence variants were identified with DADA2 (per pipeline tutorial v1.16, DADA2 v1.21.0; Callahan et al., 2016) with quality trimming at Q30 per sample as determined by FASTX-Toolkit (v0.0.14; Gordon et al., 2010). Taxonomy for all amplicons was assigned with the RDP classifier using a classifier based on the Silva 138 SSU Ref NR 99 database for 16S rRNA (Robeson et al., 2021). Relative abundance plots for taxa were generated in R using the FeatureTable package (version 0.0.11; Moore, 2020). Cluster dendrograms were generated using the pvclust R package (version 2.2.0; Suzuki et al., 2019) after double square root transformation to reduce artifacts from the large amount of ASVs that were individually relatively rare. Indicator analyses were completed using the indicspecies package in R (version 1.7.15; De Cáceres et al., 2010, 2012).

## 2.7 Magnetic measurements

Magnetic mineral measurements were made on pure iron oxide minerals used for amendments in incubations, on initial homogenized sediment prior to incubation before addition of iron amendments, and on sediments recovered at the end of the incubations. At the end of incubations, and following subsampling for DNA, sediment was recovered from slurries by centrifugation. The supernatant was discarded and sediments were frozen at -80° C under an argon headspace until they were processed further. One bottle from the three replicates was selected at random from each treatment for magnetic measurements. Initial and post-incubation sediments were freeze-dried, pulverized with a mortar and pestle and stored under an argon headspace until the next step of preparation. Powdered minerals and sediments were prepared for analysis by packing into gel capsules with quartz wool in an anaerobic glovebox. Magnetic analysis of minerals and sediments was performed on a Lakeshore 8604 Vibrating Sample Magnetometer (VSM). The VSM was used to obtain hysteresis loops and direct current demagnetization at room temperature. Data from these VSM measurements are used to characterize and quantify the mineralogy, abundance, and grain size of magnetic minerals (Fig. 4). Hysteresis loops were slope-corrected to remove antiferromagnetic, paramagnetic, and diamagnetic contributions and then saturation magnetization and coercivity were extracted for the ferromagnetic (sensu lato) components (Jackson and Solheid, 2010). The direct current demagnetization data was used to find the coercivity of remanence. Its first derivative also produced coercivity spectra, which were deconvolved using the Max Unmix software program (Maxbauer et al., 2016).

## 3 Results

### 3.1 Concentrations of Redox-Active Species by Cyclic Voltammetry

By the time cyclic voltammetry measurements were made, two hours after the core was collected, no dissolved oxygen was detected throughout the core, even at the sediment-water interface. However, at least one form of Fe(III) was detected throughout the measured interval of the sediment core, the upper 1.2 cm (Figs. 1b, A1). We detected multiple forms of Fe(III) including organic-Fe(III) aggregates and nanoparticles of different sizes (Fig A1). The Fe(III) response generally shifted to more negative applied potential with depth in the core (Fig. A1) suggesting increasingly larger particles and an aging process (Taillefert et al., 2000). Reduced sulfur species ($\Sigma S^{2-}$) were detected at 0.5 cm and deeper in the core, increasing in concentration to 78 μM at 1.0 cm. FeS was also detected at 0.5 cm depth and increased in concentration until the end of measurements at 1.2 cm (Fig. 1b). Marsh creek sediments themselves therefore supplied our incubations with iron and reduced sulfur in multiple forms and multiple oxidation states.

## 3.2 Methane Production in Incubations

Methane production was observed in all incubation bottles. The most methane, along with the highest and most consistent methane production rates, were produced in bottles amended with MMA and magnetite or hematite followed by MMA without iron amendment (Figs. 2a, 3a,b). All other treatments produced similar quantities of methane (Fig. 2a,c). The highest production rates overall were observed within the first 10 days followed by lower sporadic pulses (Fig. 3). MMA + ferrihydrite took longer to ramp up methane production compared to MMA with other iron treatments (Figs. 2a, 3c). On average, MMA + no iron bottles produced more methane than EtOH + no iron (Fig. 2a,c), however, methane production in no-iron incubations was more variable than it was with an iron amendment for both MMA and EtOH treatments (Fig. 3d). Bottles amended with EtOH produced less methane than MMA-amended bottles with the same iron treatment in all cases except with ferrihydrite amendment, in which methane production was comparable. There was also no significant difference between EtOH incubations with different iron treatments (Fig. 2c). In all EtOH-amended incubations, the highest methane production rates were observed as sporadic pulses late in the incubation period (Fig. 3). However, after the first 10 days, the trajectory of methane production was approximately the same regardless of organic carbon or iron treatment. For example, all treatments had a moderate peak in methane production on day 28 (Fig. 3). This similarity suggests that by the final two weeks of the incubation period, methane production rates were no longer related to the initial organic carbon amendments.

As $Fe^{2+}$ concentrations reflect both production by microbial iron reduction and loss from iron mineral formation, they are less sensitive as a tracer of microbial activity than methane concentrations. We therefore treat $Fe^{2+}$ concentrations as an indicator of iron reduction kinetics following the approach of previous studies (Aromokeye et al., 2018). The most net $Fe^{2+}$ production occurred in bottles with ferrihydrite amendments regardless of organic carbon treatment (Fig. 2a,c), suggesting ferrihydrite yielded the highest iron reduction rates. In EtOH bottles, magnetite and hematite yielded almost as much $Fe^{2+}$ as ferrihydrite with a similar production rate in the first 10 days (Fig. 2d). With the MMA amendment, on the other hand, iron reduction was mineral-specific. Incubations with magnetite did not ramp up $Fe^{2+}$ production until the final two weeks, and MMA + hematite incubations did not show evidence of significant iron reduction at any point during the incubation period (Fig. 2b).

## 3.3 Magnetic Mineral Formation

In bottles amended with EtOH and ferrihydrite, a progressive shift from reddish ferrihydrite to a dark gray magnetic mineral was observed over the course of the incubation (Fig. C1). To further characterize magnetic mineral formation, magnetic measurements were made. However, since such extensive transformation of ferrihydrite was not anticipated, we did not preserve subsamples of initial slurry with iron amendments for magnetic mineral characterization. Instead, initial sediments without iron amendments and each powdered iron mineral were characterized individually and used to contextualize magnetic measurements made on sediments recovered at the end of the incubations. Ferromagnetic minerals were identified and further characterized by their remanent coercivities, and saturation magnetization was used to measure ferromagnetic mineral

abundance. Magnetite-amended incubations are not discussed further in the main text because the amendment itself overwhelmed any detectable change in magnetic minerals that may have occurred during the incubation (Fig. C2, Table C1). In initial homogenized sediments, a small but measurable saturation magnetization indicated the presence of magnetic minerals (Fig. 4a). This concentration was typical of magnetic mineral concentrations measured in sediments across a wide range of conditions (Kreisler and Slotznick, *in revision*). The presence of a ferromagnetic mineral with mid-coercivity ($B_{cr}$ = 39 mT), interpreted to be magnetite, as well as a minor high coercivity component, interpreted to be hematite, were both detected (Fig. 4c). At the end of the incubation period, the saturation magnetization of sediments recovered from the iron-free treatments looked similar to initial measurements. Small increases in most other treatments are hard to deconvolute from the addition of ferromagnetic mineral amendments (Fig. 4a, hematite incubations described further in Appendix C). The iron-free, hematite, and MMA + ferrihydrite treatments all had remanent coercivities and coercivity spectra similar to the initial sediment slurry (Figs. 4b, C2). However, the EtOH + ferrihydrite treatment showed a large increase in saturation magnetization (Fig. 4a). This signal could not have resulted simply from the addition of ferrihydrite to initial sediments because ferrihydrite itself is antiferromagnetic to weakly ferromagnetic. Although the synthesized ferrihydrite analyzed contained a small ferromagnetic component, its saturation magnetization was similar to that of the hematite standard and an amended bottle with no further mineralogical changes would be comparable. This indicates that the EtOH + ferrihydrite treatment was the only one that produced significant quantities of measurable ferromagnetic minerals. A large decrease in bulk remanent coercivity was noted with EtOH + ferrihydrite treatment as well (Fig. 4b); coercivity spectra show that the predominant phase produced in the EtOH + ferrihydrite treatment is within the range for magnetite ($B_{cr} \sim 9$ mT) (Fig. 4d; Peters and Dekkers, 2003). Coercivity is not a definitive mineral identifier and other ferromagnetic minerals have overlapping coercivity ranges, specifically pyrrhotite and greigite, but the low abundance of sulfide suggests it is not either of these phases. Notably, the ferromagnetic mineral produced during EtOH + ferrihydrite incubations has a lower coercivity than the magnetite in initial sediments (Fig. 4c,d). In addition to this predominant mid-coercivity phase, there appears to be a slight increase in abundance of high-coercivity phases as well ($B_{cr}$ = 318), interpreted as hematite. A similar change is not seen in the MMA + ferrihydrite treatment, suggesting this is not simply due to abiotic aging of ferrihydrite, but could be related to microbial iron-oxidation of magnetite instead of *de novo* production.

## 3.4 Microbial Community Composition

Microbial community composition at the end of the incubations from 16S rRNA sequence data is shown in Fig. 5 for families that represented > 5% of total sequences. To facilitate more detailed comparisons, archaeal and bacterial community composition is also shown at the taxonomic level of genus in Fig. 6 for genera representing > 5% of sequences from each domain. Hierarchical clustering of 16S rRNA sequences from the end of incubations (Fig. 5a) showed a clear pattern that was aligned with carbon substrate. The same pattern was observed when just bacterial or archaeal sequences were used (Fig. B1). In all three cases, clusters are supported by bootstrap values of 100, indicating organic carbon source was the major control on both archaeal and bacterial community composition. Archaea were the dominant populations observed in all incubations.

*Bathyarchaeia* made up the largest fraction of sequences in MMA incubations, which was distinct from EtOH incubations that were dominated by *Methanosarcinaceae* (Fig. 5b).

MMA-amended incubations had Shannon Diversity Index values that were consistently higher than EtOH-amended incubations (Fig. B2), indicating MMA amendment resulted in a more diverse microbial community. Indicator Species Analysis performed at the taxonomic level of genus pointed to 68 bacterial and archaeal groups significantly associated ($p < 0.01$) with MMA at high statistic values ($> 0.95\%$). The EtOH treatments had only one indicator archaeal genus with a similarly high statistic (0.983) and p-value (0.008): *Methanosarcina* (Table B1). Iron was not found to be a strong driver of community structure with no indicator species identified for any iron treatment. In MMA incubations, three genera comprised $> 50\%$ of the sequences regardless of iron treatment: *Bathyarchaeia*, *Ca.* Omnitrophaceae, and Marine Benthic Group D/Deep Hydrothermal Vent Euryarchaeotal Group 1 (MBG-D/DHVEG-1). In EtOH incubations, on the other hand, the top three genera, *Methanosarcina*, *Syntrophotalea,* and *Bathyarchaeia*, represented almost 75% of recovered 16S rRNA sequences, in agreement with the lower diversity estimates in these treatments. We also looked for trends within archaea and bacteria genera individually (Fig. 6). Two different genera of the family *Methanosarcinaceae* were found to be indicators for carbon treatment (Fig. 6a). *Methanosarcina* was found to be an indicator for EtOH (statistic = 0.983; p-value = 0.008), while *Methanococcoides* was an indicator for MMA (statistic = 0.977; p-value = 0.004). *Methanosarcina* alone represented $> 50\%$ of archaeal sequences in EtOH + hematite and EtOH + no iron treatments and they were the largest fraction in the other iron treatments with EtOH amendment as well (Fig. 6a). The only other archaeal genera with significant representation in EtOH incubations were *Bathyarchaeia* and *Methanococcus* (Fig. 6a). However, *Bathyarchaeia* were found to be an indicator for MMA (statistic = 0.952; p-value = 0.005), not EtOH. They represented up to 51% of archaeal sequences in MMA incubations. MBG-D/DVHEG-1 were also a larger fraction of the archaeal population in MMA treatments. A group of unclassified archaea was consistently recovered from incubations with MMA treatments as well (Fig. 5b, 6b). A BLAST search indicated these unclassified Archaea were most related to *Ca.* Thorarchaeota in the Asgard superphylum (94–96% sequence similarity). None of these archaeal groups that represented the majority of archaeal sequences in MMA bottles are of the "classical" orders of methanogens (Evans et al., 2019). This was unexpected as MMA incubations produced significantly more measured methane than EtOH incubations. Classical methanogens were found at very low relative abundance, e.g., *Methanococcoides* (avg. 1.9%) and *Methanofastidiosales* (avg. 0.7%).

Bacterial assemblages were more distinct between carbon treatments than archaeal assemblages (Fig. 6b). MMA treatments resulted in high relative abundances (up to 30%) of *Ca.* Omnitrophus followed by WOR-1 (also referred to as *Saganbacteria*; (Matheus Carnevali et al., 2019), *Lentimicrobiaceae*, MSBL5 (Mediterranean Sea Brine Lake Group 5), and *Sulfurovum* (Fig. 6b). The indicator with the highest statistic for MMA treatment, a member of the *Bacteroidetes*, was not highly abundant (Table B1). But *Ca.* Omnitrophus was also a strong indicator of MMA treatment (statistic = 0.976, p-value = 0.009). In contrast, bacteria in EtOH incubations were dominated by *Syntrophotalea*, which made up $\geq 50\%$ of bacterial sequences across all iron treatments (Fig. 6b). *Anaerolineaceae* was also a substantial proportion of the community in EtOH bottles. Significant populations of Sva0081 sediment group and SEEP-SRB-1 were also found (Fig. 6b), both of which are sulfur reducers in the

family *Desulfosarcinaceae* (Dyksma et al., 2018; Skennerton et al., 2017). Only *Syntrophotalea*, however, was found to be a strong indicator genus for EtOH treatment (statistic = 0.945, p-value = 0.009).

## 4 Discussion

### 4.1 Carbon source exerts a primary control on methane production

#### 4.1.1 More methane production from non-competitive methylamine than ethanol

Our experimental design included two carbon treatments to investigate the effect of substrate competition on methane production from salt marsh microbial communities. We found that addition of a non-competitive, methylated organic carbon substrate (MMA) yields more measurable methane than a substrate that is fermented to acetate (EtOH) (Fig. 2a,c). This likely reflects an adaptation of the natural microbial assemblage to the ready availability of methylated substrates in salt marshes.
Methylamines are degradation products of common osmolytes synthesized by marsh grasses and other marsh organisms such as glycine betaine, proline betaine, choline and dimethylsulfoniopropionate (DMSP) (Cavalieri and Huang, 1981; Glob and Sørensen, 1987; King, 1988; Mulholland and Otte, 2002; Wang and Lee, 1994a). As such, methylamines are abundantly available in the sediment of salt marshes (Fitzsimons et al., 1997; Gibb et al., 1999; Lee and Olson, 1984; Oremland et al., 1982; Wang and Lee, 1994a; Yang et al., 1994). *S. alterniflora* has been shown to release methylamine to sediments (Wang
and Lee, 1994a) and, consistent with our results, active methylotrophic methanogens have previously been found in N. American salt marshes where *S. alterniflora* is native (Buckley et al., 2008; Cai et al., 2022; Franklin et al., 1988; Krause and Treude, 2021; Oremland et al., 1982; Oremland and Polcin, 1982; Parkes et al., 2012a). Our results are also consistent with the shift from hydrogenotrophic to methylotrophic methanogenesis that was observed with the invasion of *S. alterniflora* into China's coastal salt marshes where it is not native (Yuan et al., 2016, 2019).
Although our 16S rRNA sequence data reflects the microbial community at the end of the incubations rather than at the time point of highest methane production for MMA incubations (Fig. 2a,c), these bottles continued to produce methane throughout the incubation period. It was therefore somewhat surprising to find that methylotrophic methanogens were not among the more abundant genera recovered in 16S rRNA sequences in bottles incubated with MMA amendments (Fig. 5b). In fact, all representatives of the classical orders of euryarchaeotal methanogens averaged only 0.6% of recovered sequences in MMA
incubations across the four iron treatments. Instead, *Bathyarchaeia* was the most represented among the archaea by far (averaged 13% of all sequence reads in MMA incubations). As a class, *Bathyarchaeia* are found abundantly in anoxic marine sediments and terrestrial soils (Fillol et al., 2016; Zhou et al., 2018) where they are thought to be important methylotrophic acetogens that degrade a variety of organic carbon substrates (Farag et al., 2020; Hou et al., 2023). It has been proposed that methylotrophic methanogenesis may also be possible among the *Bathyarchaeia* based on genomic data (Evans et al., 2015)
but it has not been shown experimentally, and a recent study of existing metagenomes indicated that methanogenesis would

not be widespread among them (Hou et al., 2023). Therefore, it is not likely that *Bathyarchaeia* produce methane in MMA-amended incubations although it is possible.

A more likely source of the methane production in MMA incubations is a population of classical methylotrophic methanogens that is highly active, despite representing a small proportion of the sequences recovered at the end of incubations. A small population of methanogens with high activity has been described previously in salt marsh sediments (Cai et al., 2022). Of the sequences we recovered from established methanogens, members of the genus *Methanococcoides* were most abundant (Fig. 6a). *Methanococcoides* have been identified in other studies of salt marsh sediments (Jameson et al., 2019; Jones et al., 2019; Munson et al., 1997) and they are known to utilize methylamines and other methylated compounds such as choline and glycine betaine, but not $H_2$ or acetate (L'Haridon et al., 2014; Liang et al., 2022; Singh et al., 2005; Watkins et al., 2014). A similarly high relative abundance of *Bathyarchaeia* and smaller abundance of classical methanogens, specifically *Methanococcoides*, have also been reported in seagrass meadow sediment where methylated compounds fuel methane production via methylotrophic methanogenesis (Hall et al., 2024; Schorn et al., 2022).

After *Methanococcoides, Methanofastidiosales* were the next most abundant of the established methanogens in MMA-amended incubations (Fig. 6a). *Methanofastidiosales* are restricted to methanogenesis through methylated thiol reduction (Nobu et al., 2016). Methanethiol, which can be formed as a degradation product of DMSP and methionine or by methylation of sulfide (Lomans et al., 2002), is also commonly available in salt marsh sediments (Carrión et al., 2019; Dévai and Delaune, 1995; Kiene and Visscher, 1987) and was likely to have been naturally present in the sediments used for our incubations. Notably, *Methanofastidiosales* were found along with *Methanococcoides* and *Bathyarchaeia* in methanic sediments from seagrass meadows (Schorn et al., 2022). While the proportion of methanogen 16S rRNA sequences was lower overall in MMA treatments compared to EtOH treatments at the end of the incubations (Figs. 5b, 6a), the higher methane production rates observed with MMA likely indicates that a small population of salt marsh methanogens had high potential activity. They could have been primed to use non-competitive methylated substrates, such as methylamines and methanethiol, that are not accessible to sulfate- or iron-reducing bacteria. Methanogens in our incubations could also have been supported by the products of organic matter degradation by *Bathyarchaeia*, as has been reported previously in other iron-rich, sulfate-poor sediments (Ruiz-Blas et al., 2024).

**4.1.2 Ethanol yields a larger share of methanogens and potential concurrent methane oxidation**

Incubations amended with EtOH also produced methane throughout the incubations, although less and later than with MMA amendments (Figs. 2, 3). EtOH would not directly support methanogens, but it can be fermented to methanogenic substrates by other members of the microbial community (Du et al., 2021; Schink, 1997). The lag in methane production in EtOH incubations relative to MMA incubations was likely related to this requirement that ethanol is converted to acetate or $H_2$ by bacteria before it could be utilized. The two most abundant bacteria in incubations amended with EtOH were *Syntrophotaleaceae* and *Anaerolineaceae* (Figs. 5b, 6b). Both of these bacterial genera are likely involved in organic carbon fermentation (Hackmann, 2024; McIlroy et al., 2017). *Syntrophotalea* was the most abundant bacterial genus by far (Fig. 6b)

and was second most abundant among all taxa recovered from EtOH incubations (12%–17%). Isolates of the genus *Syntrophotalea* (formerly *Pelobacter*) have been shown to ferment ethanol in the presence of a $H_2$-scavenging methanogen and to produce acetate among their fermentation products (Eichler and Schink, 1985; Schink, 1984, 2006). More recently, a member of the *Syntrophotalea* was shown to transfer formate to a *Methanococcus* isolate in a mutualistic, methane-producing relationship (Day et al., 2022). *Anaerolineaceae*, although not as abundant, also likely contributed to fermentation in EtOH incubations. They have been identified as primary fermenters capable of degrading carbohydrates, proteinaceous material or alkanes in anaerobic digesters (Bovio-Winkler et al., 2020; Liang et al., 2015; McIlroy et al., 2017). In a few instances, they have also been found associated with acetoclastic methanogens from the *Methanothrix* (formerly *Methanosaeta*) genus (Liang et al., 2015; McIlroy et al., 2017).

Even though measured methane production was lower overall (Fig. 2c), and acetate and $H_2$ are competitive substrates that can be consumed by both methanogen and non-methanogen members of the microbial community, classical methanogens represented a much larger share of the 16S rRNA sequences recovered at the end of EtOH incubations (Fig. 5b). *Methanosarcina* was the dominant microbial genus, representing 15% to 25% of sequences recovered from EtOH incubations. Although *Methanosarcina* are capable of hydrogenotrophic, acetoclastic and methylotrophic methanogenesis, their ability to use acetate at high concentrations (Stams et al., 2019) was likely related to their proliferation in EtOH-amended incubations. *Methanococcus*, a genus of hydrogenotrophic methanogen, represented an additional 4 to 8% of total methanogen sequences in our EtOH incubations (Fig. 6a). Their abundance allows for the possibility that a mutualistic relationship such as the one reported for *Syntrophotalea* and *Methanococcus* in an earlier study (Day et al., 2022), may have supported *Methanococcus* in our incubations as well. Although we did not find acetoclastic *Methanothrix* among the methanogens in our EtOH incubations, it is possible that a similar cooperative interaction that was described between *Anaerolineaceae* and *Methanothrix* (Liang et al., 2015; McIlroy et al., 2017) formed between *Anaerolineaceae* and an acetoclastic *Methanosarcina* in our incubations.

The larger proportion of methanogens but lower methane production observed in EtOH incubations relative to MMA incubations could result from a less active population of methanogens than in the MMA-amended bottles. However, an alternative explanation is that our measurements of methane concentrations underestimated methane production due to methane oxidation in EtOH incubations. Concurrent methane production and oxidation has been reported in salt marsh sediments (Krause et al., 2023, 2025; Krause and Treude, 2021; Segarra et al., 2013) as well in other coastal settings (Krause et al., 2023; Xiao et al., 2017, 2018). Methane oxidation is suggested by our identification of small numbers of ANME-3. Although ANME-3 were a small minority of sequences (included in "Other" in Fig. 6a), they were most represented in EtOH incubations with ferrihydrite, magnetite and hematite (1%, 0.3% and 0.8%, respectively). We also detected the presence of the sulfate-reducing partner of anaerobic methanotrophs, SEEP-SRB1 (Schreiber et al., 2010; Skennerton et al., 2017), in all EtOH-amended incubations (Fig. 6b). Although sulfate-free artificial seawater was used to make up our sediment slurry, natural reduced sulfur of multiple forms was identified in the marsh creek sediments (Fig. 1b). Although we did not measure sulfur species during the incubations and reduced sulfur native to the sediments would have been diluted by our use of sulfate-free artificial seawater, sulfur cycling is suggested by the presence of sulfur oxidizing *Thiotrichaceae* (Ravin et al., 2023) in

addition to the sulfate-reducing *Desulfosarcinaceae* (Fig. 5b). It is also possible that sulfate-reducing bacteria were reducing Fe(III) in our incubations (Coleman et al., 1993; Lovley et al., 1993), or anaerobic methane oxidation could be coupled to Fe(III) reduction (Cai et al., 2018; Ettwig et al., 2016; Yu et al., 2022), which has been observed previously in salt marsh sediments (Krause and Treude, 2021; Segarra et al., 2013). Overall, a picture emerges of a complex web of interspecies interactions mediated by multiple indirect electron shuttles and redox-active species in EtOH incubations.

## 4.2 Conductive minerals promote increased methylotrophic methane production

We expected EtOH incubations to reveal the potential for IET to enhance methane production by salt marsh microbial communities based on the results of previous studies. Many studies of co-cultures and enrichments in which IET was found use ethanol to enrich the bacterial and methanogen partners and to support methane production (Kato et al., 2012; Rotaru et al., 2019, 2014, 2018; Tang et al., 2016). Transfer of electrons between methanogens and electroactive bacteria has been shown through both magnetite and hematite (Aromokeye et al., 2018; Kato et al., 2012; Zhou et al., 2014). But (semi)conductive iron minerals have also been shown to enhance the activity of acetoclastic methanogens themselves, without a bacterial partner (Fu et al., 2019; Wang et al., 2020; Xiao et al., 2021). The typical methanogen involved in either pathway that accelerates methane production is from the metabolically versatile genus *Methanosarcina* or the acetoclastic genus *Methanothrix* (Rotaru et al., 2021; Xiao et al., 2021). Although we did not detect *Methanothrix*, *Methanosarcina* was abundantly represented in our EtOH incubations (Fig. 5b). However, we did not observe any definitive increase in methane production in the presence of either magnetite or hematite relative to no iron amendments in EtOH incubations (Fig. 2d). This suggests IET was not active in EtOH incubations. Genera of well-known electroactive bacteria (Garber et al., 2024; Lovley and Holmes, 2022) were also not detected in EtOH incubations.

Instead, the most methane produced and highest methane production rates that we observed were found in MMA + magnetite incubations followed by MMA + hematite incubations in the first 10 days (Figs. 2,3). Although methane production with no iron amendment was variable, the average among replicates was lower than that with magnetite or hematite amendments and methane production with MMA + ferrihydrite was significantly lower. This pattern of higher methane production rates with magnetite and hematite than with ferrihydrite or no iron has been reported previously in rice paddy soil microbial enrichments (Kato et al., 2012; Zhou et al., 2014). However, the most abundant methanogens in MMA incubations with these iron treatments were neither *Methanosarcina* nor *Methanothrix*. A few other methanogenic genera have been found capable of participating in IET (e.g., *Methanobacterium*; (Zheng et al., 2020, 2021), suggesting the full diversity of IET-capable methanogens is not yet known. Methylotrophic methanogens, such as those that were most abundant in our MMA incubations (Fig. 6a), have not been identified as capable of IET to date. Still, *Methanococcoides* are a member of *Methanosarcinales*, an order with other IET-capable genera. Some *Methanococcoides* species have been found with cytochromes (Jussofie and Gottschalk, 1986; Saunders et al., 2005) that could facilitate extracellular electron transfer.

The bacterial partner in IET interactions is more variable and therefore more difficult to predict. In studies of IET-active co-cultures and enrichments, *Geobacter* is often the syntrophic partner of methanogens and can be found in high abundance, but

not always (Deng et al., 2023; Holmes et al., 2017; Kato et al., 2012; Liu et al., 2015; Rotaru et al., 2014; Zhou et al., 2014). We did not recover 16S rRNA sequences belonging to *Geobacter* or other bacterial genera that are found in syntrophic association with methanogens (e.g., *Syntrophomonas*; Yuan et al., 2020; Zhao et al., 2018) or *Rhodoferax* ; Yee and Rotaru, 2020; Zhao et al., 2020). Instead, the most abundant bacterial genus was *Ca*. Omnitrophus (Fig. 6b). *Ca*. Omnitrophus has widespread distribution in the environment, but they have remained relatively uncharacterized. Members of the *Ca. Omnitrophus* genus have been associated with metal-contaminated groundwater (Bärenstrauch et al., 2022) and magnetite production in the form of magnetosomes .(Kolinko et al., 2016; Lin et al., 2018). In the class *Omnitrophia,* metagenomic evidence indicates potential to produce cytochromes for use in dissimilatory metal reduction and conductive pili (Perez-Molphe-Montoya et al., 2022; Seymour et al., 2023). These features allow for the possibility that *Ca*. Omnitrophus are the electroactive syntrophic partner of *Methanococcoides* in our MMA incubations. However, *Ca*. Omnitrophus was also found abundantly in MMA incubations with no iron and ferrihydrite treatments (Fig. 6b), and these incubations did not appear to have active IET. The other bacterial genera with the highest abundances, WOR-1, or *Saganbacteria*, *Lentimicrobiaceae*, MSBL5, and *Sulfurovum* (Fig. 6b) are less likely to be the electroactive bacterial partner. *Lentimicrobiaceae* are thought to be fermenters (Sun et al., 2016), *Sulfurovum* are sulfur reducing bacteria (Li et al., 2023; Wang et al., 2023), and metabolic capabilities and biogeochemical roles of *Saganbacteria*, and MBSL5 are largely unknown. Like *Ca.* Omnitrophus, the abundances of these genera are also not distinctly different between bottles with magnetite and hematite amendments vs. bottles with ferrihydrite and no iron amendments. Despite the distinctly higher methane production rates early on, the microbial communities in MMA + magnetite and MMA + hematite incubations were not significantly different from MMA + no iron and MMA + ferrihydrite by the end of the incubations. The similar bacterial assemblages in MMA incubations may reflect similar methane production rates at the end of the incubations (Fig. 3) or they could be explained by the microbial community responding to the small amount of magnetite naturally present in salt marsh sediments (Fig. 4).

## 4.3 Magnetite formation from ferrihydrite with ethanol supplementation

Despite general similarities in methane production and microbial community composition in all incubations with the same organic carbon amendment, large quantities of magnetic minerals were produced in only one of our incubations: EtOH + ferrihydrite (Figs. 4, C1). Coercivity, remanent coercivity, and coercivity spectra suggest the mineral is magnetite, and that the magnetite produced in EtOH + ferrihydrite incubations was different from that natively present in marsh creek sediments (Fig. 4c,d). The low coercivity suggests very fine-grained magnetite (above 25 nm but probably not over 80 nm; (Dunlop, 1973; Dunlop and Ozdemir, 2011; Enkin and Dunlop, 1987). It is also lower coercivity than seen in magnetosomes suggesting extracellular precipitation (Bai et al., 2022; Egli, 2004; Xue et al., 2024). Such a transformation of ferrihydrite is not surprising considering that iron-reducing bacteria are known to facilitate the conversion of poorly crystalline Fe(III) minerals to more crystalline iron-bearing minerals such as magnetite (Benner et al., 2002; Hansel et al., 2003; Zachara et al., 2002). However, no detectable magnetite was formed from ferrihydrite during incubation with MMA (Fig. 4a) even though $Fe^{2+}$ concentrations were elevated during the incubation indicating that iron reduction occurred (Fig. 2b), and there was a small delay in methane

production, indicating that iron reduction was competing with methanogenesis early in the incubations (Fig. 2a). Although the mineral fate of Fe(II) produced by iron reducers has been tied to the rate of iron reduction (Kappler et al., 2023), incubations with EtOH + ferrihydrite, MMA + ferrihydrite and EtOH + magnetite treatments all produced similar amounts of net $Fe^{2+}$ (Fig. 2b,d) suggesting similar iron reduction rates. Microbial community composition did not provide strong clues as to the identity of the bacteria responsible for magnetite formation either. There was no significant difference between the bacterial community in the EtOH + ferrihydrite incubation and other EtOH treatments (Fig. 6b). No well-known dissimilatory iron reducers (or magnetotactic bacteria) were among the bacterial genera that we recovered from EtOH incubations (Bazylinski et al., 2013; Kappler et al., 2021; Simmons and Edwards, 2007a; Zhao et al., 2025), with the exception of *Ca.* Omnitrophus. In contrast to MMA incubations, however, only a small fraction of sequences in EtOH incubations were from *Ca.* Omnitrophus. Although the genus *Ca.* Omnitrophus does include magnetotactic bacteria (Kolinko et al., 2016; Perez-Molphe-Montoya et al., 2022), they were much less abundant in EtOH incubations than MMA incubations. There is a possibility that bacteria of the genus *Methanosarcina*, which were highly abundant in EtOH incubations, are directly involved in iron reduction and ferromagnetic mineral formation as cultures of *Methanosarcina barkeri* have been shown capable of ferrihydrite reduction (Bond and Lovley, 2002; Sivan et al., 2016; Yu et al., 2022).

In rice paddy soils, transformation of ferrihydrite to magnetite results in an eventual enhancement in methane production (Tang et al., 2016; Zhuang et al., 2015, 2019), but the impact of magnetite formation on methane production has not been characterized yet in salt marshes. We did not observe a similar increase in methane production following magnetite formation in EtOH + ferrihydrite incubations as that observed in rice paddy incubations, at least on the timescale of our 34-day incubations. However, this may result from a mismatch between the more prolific methylotrophic methane producers in our salt marsh incubations and the formation of magnetite, which only occurred in EtOH incubations. Although we did not detect magnetite formation in MMA + ferrihydrite incubations, the widespread presence of crystalline iron minerals at anoxic and euxinic depths resulting from transformation of amorphous iron minerals has been reported previously (Kostka and Luther, 1995; Taillefert et al., 2007). Separate populations of iron reducing bacteria accessing different carbon substrates may be involved in magnetite formation and IET in salt marshes. The activity of both would influence rates and quantities of methane production by methylotrophic methanogens.

**5 Conclusions**

In this study, we hypothesized that the noncompetitive, methylated substrate MMA would yield more methane production than EtOH without the addition of iron oxide minerals, but that the acetoclastic methanogens that we expected to find in EtOH incubations would respond to the presence of (semi)conductive iron oxides with increased methane production while methylotrophic methanogens in MMA incubations would not. Instead, we found that a noncompetitive substrate and the presence of conductive minerals worked in concert to yield the highest methane production from salt marsh microbial communities. Methanogens in salt marshes therefore can employ multiple simultaneous strategies to circumvent the redox

ladder. Adding a methylated organic carbon substrate had a stronger effect on measured methane production and on the microbial community than the presence of conductive minerals. This suggests that it is not just the presence of sulfate in polyhaline salt marshes that controls methane production, but also the contribution of methyl-containing osmolytes from marsh plants and animals that support methane production in higher-salinity coastal environments. The transformation of ferrihydrite

to magnetite and the persistent presence of crystalline iron minerals in anoxic zones of salt marsh sediments could further enhance methane production in salt marshes. The importance of these multiple factors and their inevitable variability across space, depth and seasons are likely to contribute to the heterogeneity that has been observed in methane emissions from salt marshes.

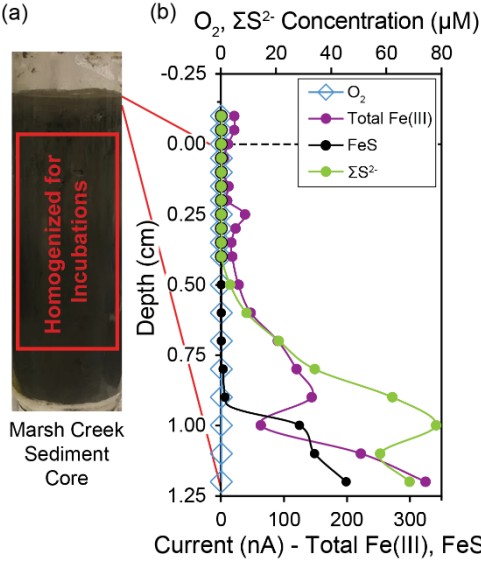

**Figure 1: Redox conditions and sampling depths in the marsh creek sediment core. (a) Photo of 22.5 cm sediment core before extrusion with interval of sediment used for incubations highlighted. (b) Microelectrode profiles of oxygen, iron and sulfur species with depth in the sediment core as measured by cyclic voltammetry. Total Fe(III) is        the summed peak areas for Fe(III)$_{(aq)}$, Fe(III)$_{(s)}$ nanoparticles of multiple sizes, and Fe(III) with organic ligands. Total Fe(III) and FeS are plotted with relative units. Traces of FeS were detected beginning at 0.5 cm. O$_2$, Fe$^{2+}$, and HS$^-$ were not detected in the sediment core or overlying water. The dashed**

**line at 0 cm represents the sediment water interface and negative depths represent height in overlying water.**


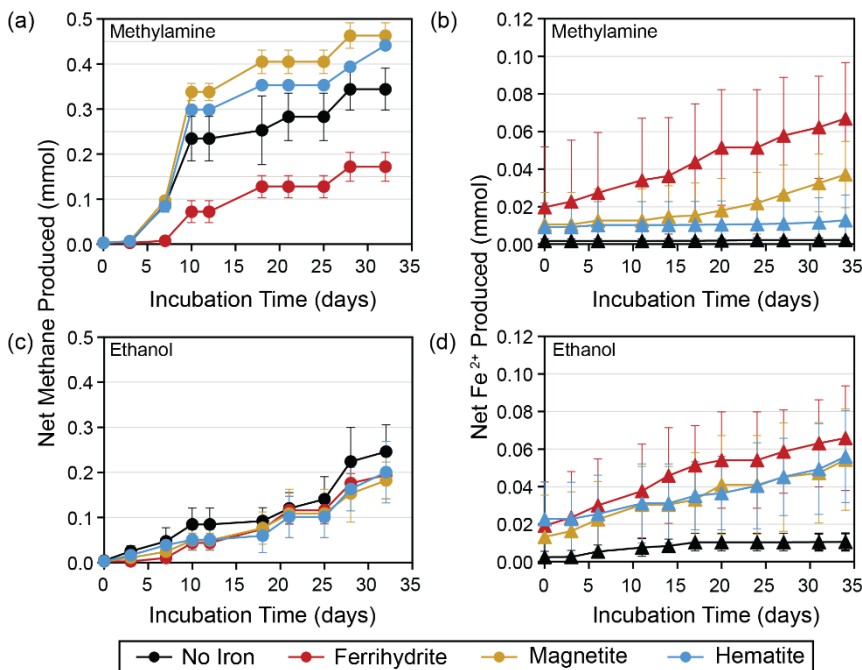

Figure 2: (a) Net methane and (b) $Fe^{2+}$ produced with methylamine amendments and all iron treatments averaged over three replicates. Similar plots for (c) net methane and (d) $Fe^{2+}$ produced with ethanol amendments are also averaged over three replicates. Error bars represent the standard deviation from the replicates' average value.





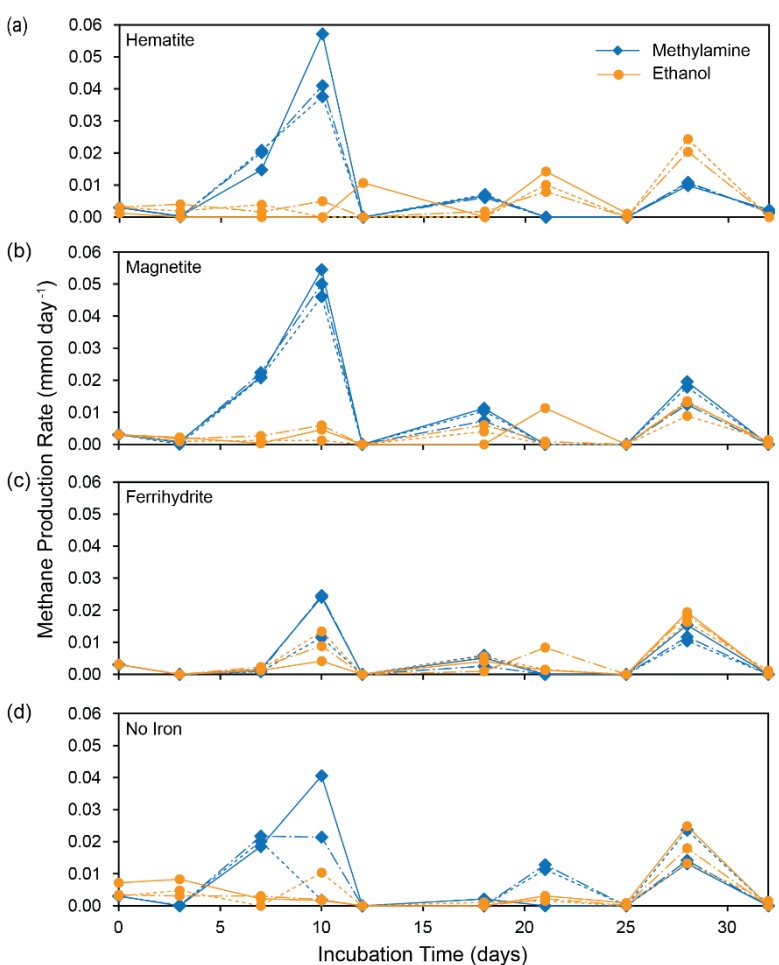

**Figure 3: Methane production rates for each replicate incubation bottle shown for MMA incubations (blue) and EtOH incubations (orange) separated by iron treatment.**




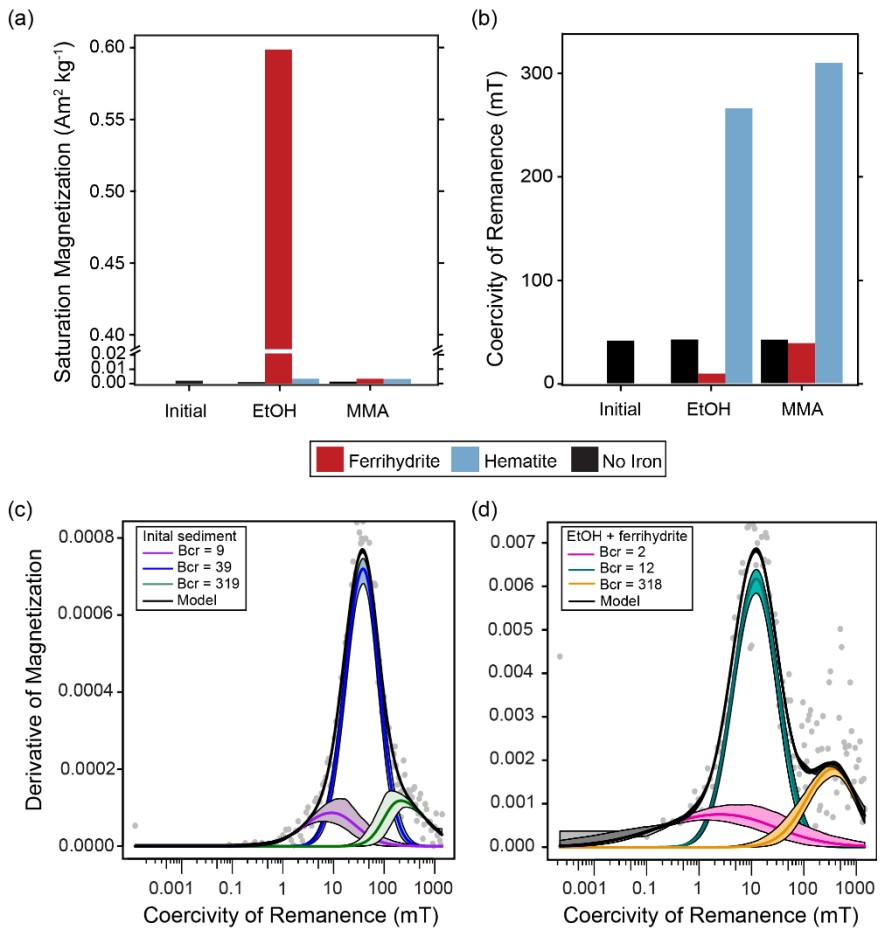

**Figure 4: Ferromagnetic mineral measurements of initial homogenized sediment used in all incubations and sediment recovered from all amended incubations at the end of the incubation period. (a) Abundance of ferromagnetic minerals as indicated by saturation magnetization ($M_s$; $Am^2\ kg^{-1}$ of dried sediment sample). (b) Mineral identification summarized by coercivity of remanence ($B_{cr}$; mT). Examples of MAX UnMix data for (c) initial sediment with no iron mineral amendment and (d) EtOH + ferrihydrite sediment at the end of the incubation period. Data from sediments amended with magnetite are not shown as magnetite amendment itself overwhelmed any signals of magnetite formation or loss.**




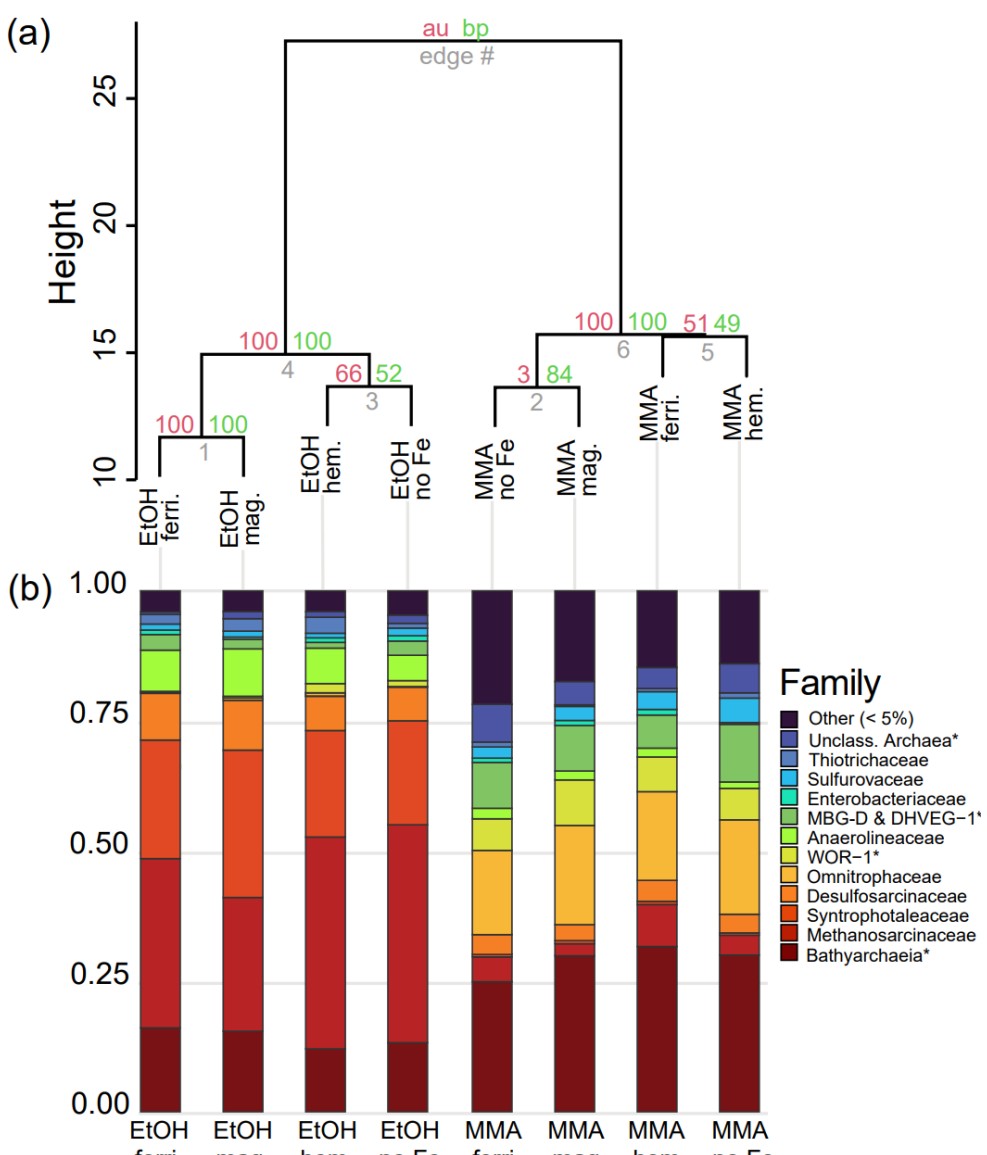

**Figure 5: Microbial community composition from 16S rRNA gene diversity at the end of the incubation period. (a) Cluster dendrogram of all gene sequences recovered with a universal 16S rRNA primer. Approximately unbiased probability (AU, red) and bootstrap probability (BP, green) are shown as percentages at nodes. (b) Relative abundances of the most abundant microbial groups classified at the level of family. ASVs with less than ten counts were grouped into the "Other" category. *Archaea and bacteria that could only be classified at higher taxonomic levels are noted with an asterisk.**



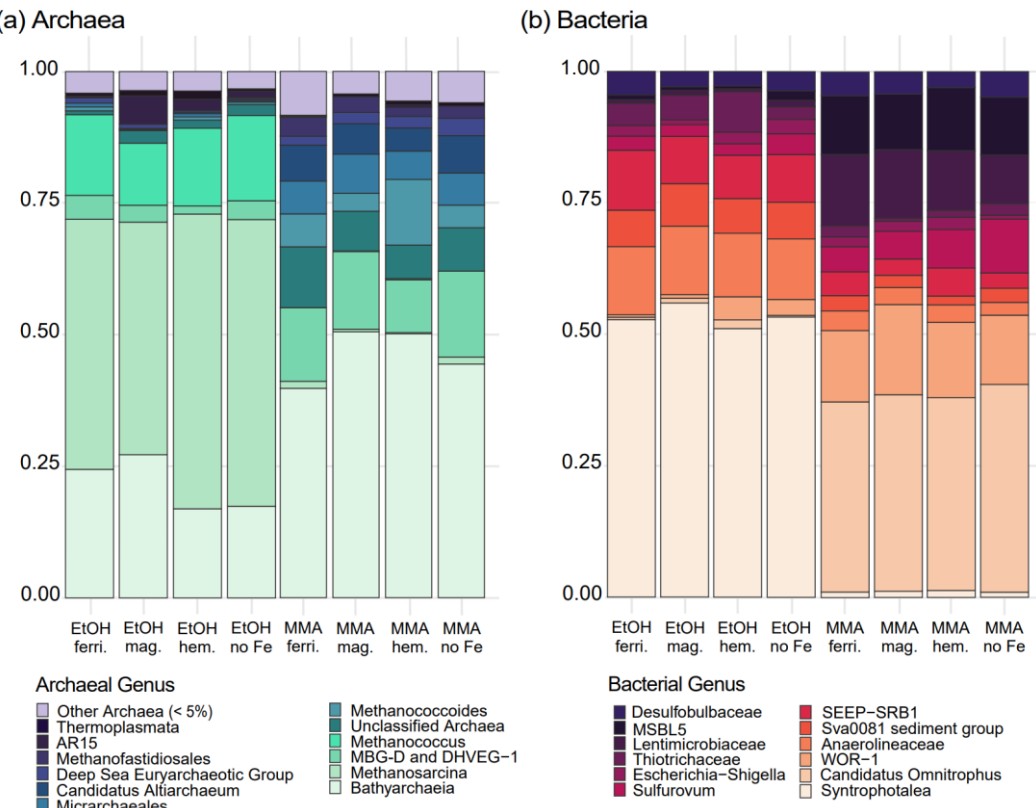

**Figure 6. Most abundant bacterial (a) and archaeal (b) genera from each incubation group from the universal 16S primer dataset. Bacterial data includes ASVs with 10 or more counts, and archaeal data includes ASVs that had 2 or more counts. Cluster dendrograms of archaeal and bacterial sequences are included separately in Appendix Figure B1.**

**Appendix A: Extended Cyclic Voltammetry Results**

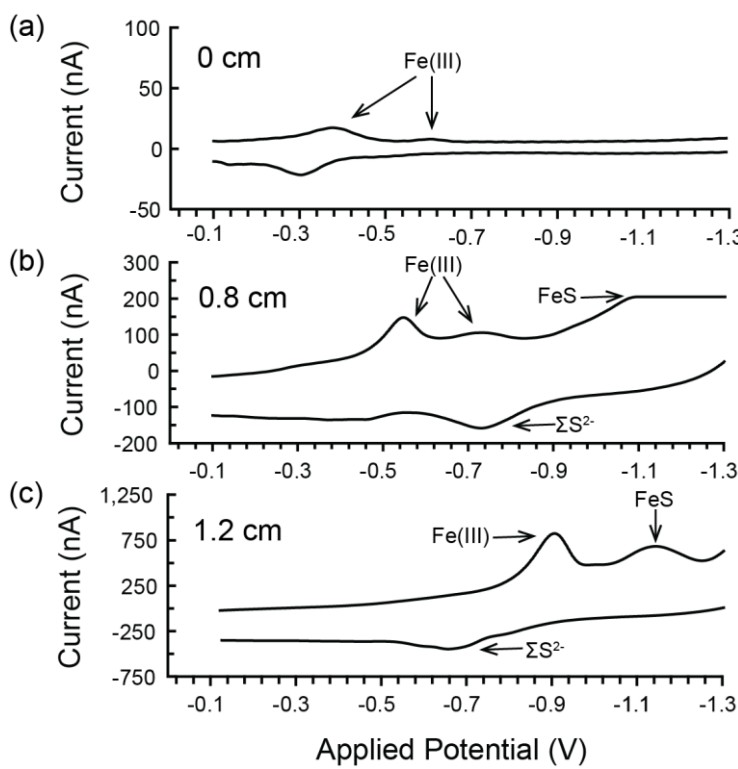

**Figure A1: Example voltammograms from depths of distinct redox conditions for (a) the sediment-water interface at 0 cm, (b) 0.8 cm depth and (c) 1.2 cm depth in the sediment. Traces of FeS are detected beginning at 0.8 cm, however, the small peak shown (b) is almost entirely off scale.**




## Appendix B: Extended Results from DNA Analysis

| Indicator classes for | Genus | Statistic | p-value |
|---|---|---|---|
| Ethanol | Methanosarcina | 0.983 | 0.0083 |
| | Methanococcus | 0.945 | 0.0094 |
| | Syntrophotalea | 0.937 | 0.0094 |
| | SCGC AAA286-E23 | 0.933 | 0.0330 |
| | Syntrophotalea | 0.924 | 0.0063 |
| | | | |
| Methylamine | Bacteroidetes BD2-2 | 0.981 | 0.0089 |
| | Methanococcoides | 0.977 | 0.0040 |
| | Candidatus Omnitrophus | 0.976 | 0.0089 |
| | Rs-M47 | 0.974 | 0.0089 |
| | MSBL5 | 0.974 | 0.0060 |
| | Roseimarinus | 0.974 | 0.0089 |
| | SBR1031 | 0.971 | 0.0065 |
| | 661239 | 0.970 | 0.0065 |
| | DSEG | 0.961 | 0.0038 |
| | Marinimicrobia (SAR406 clade) | 0.961 | 0.0089 |
| | Desulfatiglans | 0.961 | 0.0089 |
| | Unclassified Archaea | 0.959 | 0.0089 |
| | Bathyarchaeia | 0.952 | 0.0054 |
| | CK-2C2-2 | 0.951 | 0.0089 |

**Table B1. Indicator genera for carbon and iron treatments based on 16S rRNA classified at the level of genus. Because of the large number of indicators found for methylamine treatment, only genera with a statistic value of 0.95 or greater are included for this category.**


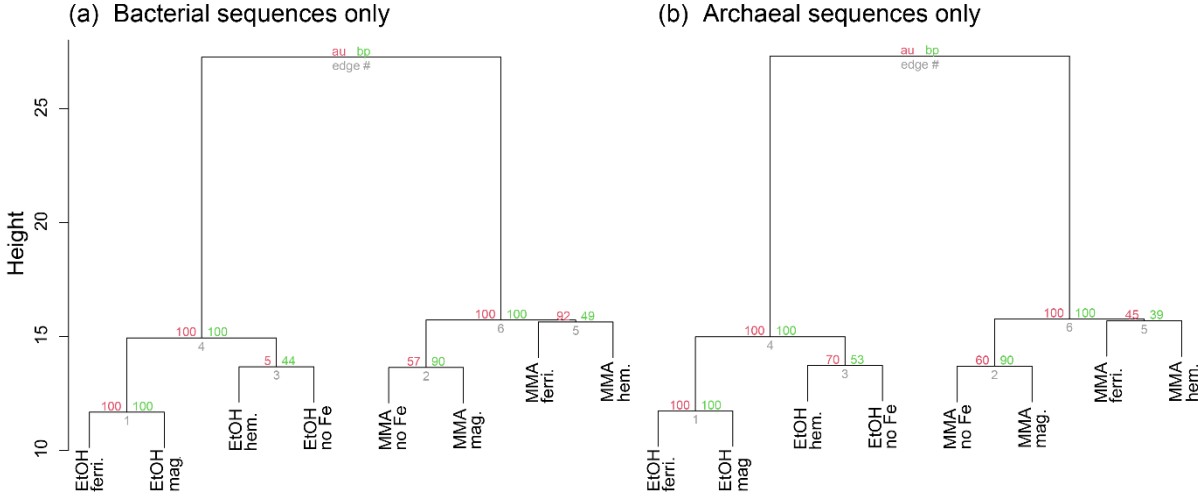


**Figure B1: Cluster dendrograms for DNA-based 16S rRNA gene diversity sorted only to the (a) Domain Bacteria and the (b) Domain Archaea. Approximately unbiased (AU) p-values are in red (in percentage), and bootstrap probability (BP) values are shown in green.**

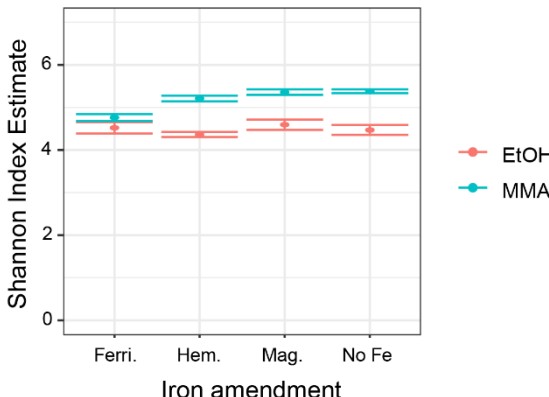

**Figure B2: Shannon Diversity Index values for microbial assemblages at the end of the incubation period.**

## Appendix C: Extended Results and Discussion of Magnetics Analyses

**Sediments from Incubations amended with Hematite**. A higher saturation magnetization was observed in the MMA + hematite treatment compared to the EtOH + hematite treatment. Coercivity spectra are similar between the two treatments with two clear magnetic phases (hematite and magnetite), but the relative ratios of magnetite and hematite differ slightly. The ratio is 43:57 in the EtOH + hematite treatment versus 37:63 in the MMA + hematite treatment. This is reflected in the slightly lower coercivity values in the EtOH + hematite treatment than the MMA + hematite treatment (Fig. 4b, Table C1).


| Sample Description | Saturation Magnetization ($M_s$, Am$^2$ kg$^{-1}$) | Saturation Remanent Magnetization ($M_{rs}$, Am$^2$ kg$^{-1}$) | Coercivity ($B_c$, mT) | Coercivity of Remanence ($B_{cr}$, mT) |
|---|---|---|---|---|
| Ferrihydrite (synthesized) | 0.340179 | 0.000145813 | 0.606429 | 95.2 |
| Magnetite (standard) | 88.4135 | 9.0119 | 11.8811 | 36.324 |
| Hematite (standard) | 0.369133 | 0.122205 | 381.008 | 650.458 |
| Sediment + no amendments | 0.001858 | 0.000415673 | 15.0399 | 41.1813 |
| MMA + no iron | 0.001299 | 0.000322783 | 15.9069 | 42.0022 |
| MMA + ferrihydrite | 0.003299 | 0.000564935 | 15.1455 | 38.6557 |
| MMA + magnetite | 3.61366 | 0.225262 | 6.91942 | 22.5718 |
| MMA + hematite | 0.00305543 | 0.00122488 | 50.5933 | 309.928 |
| EtOH + no iron | 0.001077 | 0.000356518 | 19.5832 | 42.3108 |
| EtOH + ferrihydrite | 0.598696 | 0.0057064 | 0.540556 | 9.24095 |
| EtOH + magnetite | 2.69794 | 0.182766 | 7.7745 | 30.1945 |
| EtOH + hematite | 0.003332 | 0.00128976 | 47.9657 | 265.949 |

**Table C1. Magnetics measurement results for iron minerals and all incubations at the beginning of the incubation periods (sediment + no amendments) and at the end of the incubation period (all other incubations).**

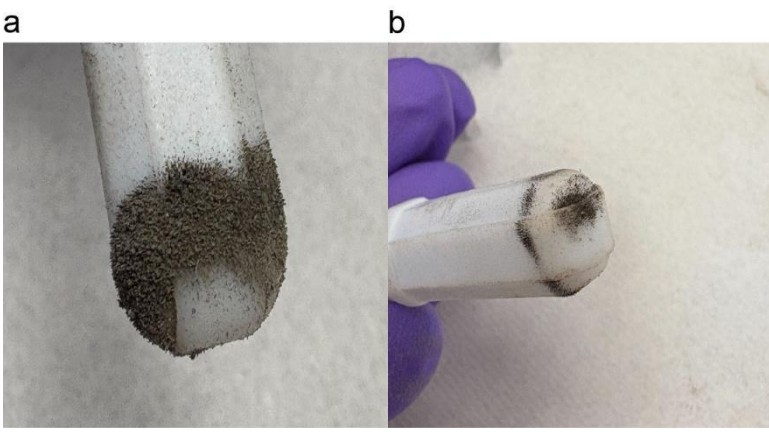


**Figure C1: Photo of ferromagnetic minerals pulled out of freeze-dried sediment with a magnetic stirbar. These sediments were recovered from EtOH + ferrihydrite incubations at the end of the incubation period. The magnetic mineral sticking to the stirbar is likely magnetite and/or greigite. (a) Stirbar with magnetic mineral and sediments. (b) Stirbar after sediments had been tapped off, more selectively leaving the magnetic mineral attached.**


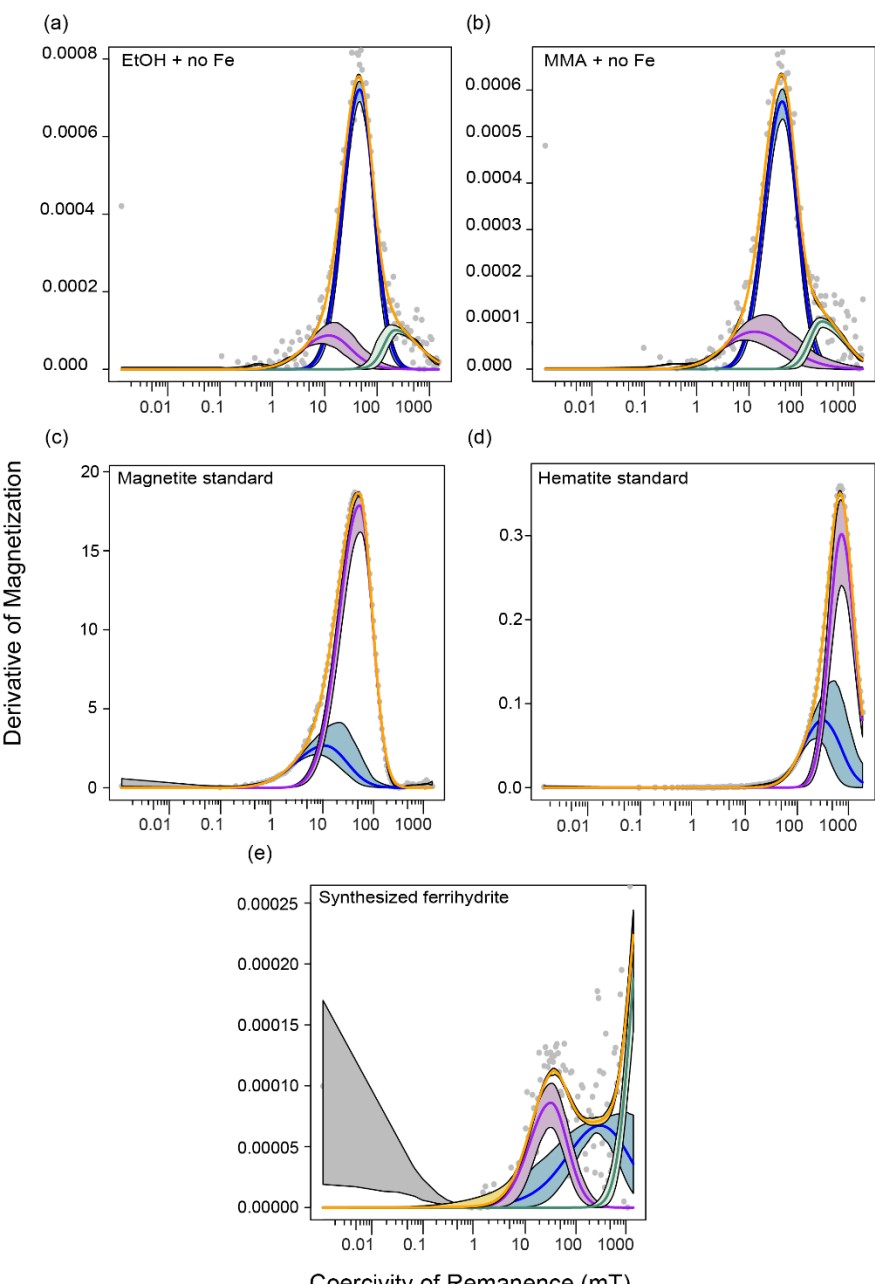

**Figure C2: MAX UnMix coercivity spectra of (a) EtOH with no iron amendment at the end of the incubation period, (b) MMA with no iron amendment at the end of the incubation period, (c) the magnetite standard used in all magnetite-containing incubations, (d) the hematite standard used in all hematite-containing incubations and (e) the synthesized ferrihydrite used in all ferrihydrite-containing incubations. Because ferrihydrite is only weakly ferromagnetic, there is high noise in the data making it difficult to deconvolve a peak.**


## Code Availability

A Rmd file used to generate all DNA figures and data tables can be found at: https://hansonlabgit.dbi.udel.edu/hanson/block_et_al_manuscript.git.

## Data Availability

Amplicon sequence data is available at: https://www.ncbi.nlm.nih.gov/bioproject/PRJNA1200716/.

## Author Contribution

This project was conceptualized by S.R.S.W. with the input of K.R.B. and A.A. Development of methodology and investigation was done by K.R.B, A.A and S.R.S.W. All authors contributed to data curation and formal analyses. Data visualization was done by K.R.B., S.R.S.W., T.E.H. and S.P.S. Original draft preparation was done by K.B. and S.R.S.W. All authors contributed to reviewing and editing the manuscript. Funding acquisition, project administration and supervision were done by S.R.S.W.

## Competing Interests

The authors declare that they have no conflict of interest.

## Acknowledgements

Undergraduate interns participating in an NSF-supported Marine Sciences REU program over multiple years contributed to the development of this project: Sarah Bartoloni in 2019, Ethan Goulart in 2021 and Samantha Reynolds in 2024, as well as an undergraduate intern participating in the UD Summer Scholars program, Matthew Hicks, in 2021. Particular thanks is owed to S. Reynolds for her contributions and discussions during the writing of this manuscript. We also thank lab members Justin Guider, Sawyer Hilt and Leland Wood, for their assistance with field work, Andrew Wozniak for the use of his GC-FID instrument and Jennifer Biddle for her advice and suggestions during the years of iterative improvement. Our manuscript was significantly improved by helpful discussions about cyclic voltammetry data interpretation with Mustafa Yücel.

## Financial Support

The development of this project was supported by NSF OCE-2342979 to S.R.S.W. The participation of A.A. and previous undergraduate interns were made possible by the UD Marine Sciences Research Experiences for Undergraduates (REU)

program administered by Joanna York (NSF OCE-2051069). K.B. was supported by NSF OCE-2342979 as well as University

of Delaware Graduate Scholars Award.

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
