# Peer review of "Influence of Carbon Source and Iron Oxide Minerals on Methane Production and Magnetic Mineral Formation in Salt Marsh Sediments"

_EGUsphere, 2025_

## Author Response (AR1)

We appreciate the opportunity to submit a revised manuscript and would like to thank our two reviewers and associate editor for their valuable suggestions and recommendations. Their comments have improved both the quality and clarity of our manuscript. Below, we have provided detailed, point-by-point responses to all comments and suggestions. For each comment, we first reproduce the original reviewer/editor remark (italicized), followed by our response and a description of the corresponding changes made in the revised manuscript. Line numbers refer to the original version of the manuscript.

**Reviewer #1**

The manuscript written by Block et al., investigated the response of microbial communities and geochemistry in salt marsh sediments that were exposed to a combination of added carbon substrates (i.e., mono-methylamine (MMA) and ethanol) and various iron (III) bearing minerals. The authors did this by preparing a sediment slurry with salt marsh sediments and sulfate-free artificial seawater which was transferred into replicate microcosms and added the various combinations of carbon substrate and iron bearing minerals. The authors tracked the production of methane and changes in the iron bearing minerals over time, as well as, compared the bacterial and archaeal communities between treatments. The authors found that MMA along with magnetite or hematite led to the highest amount of methane production from the salt marsh sediment slurry, suggesting that the presence of MMA and iron bearing minerals enhances methane production. Whereas incubations amended with ethanol yielded less methane than the microcosms treated with MMA, however, transformation of hematite to magnetite was observed. The authors also found genomic evidence of anaerobic methanotrophs and hinted at potential cryptic methane cycling. The main conclusion the authors make is that salt marsh sediment that contain microbially-mediated magnetite could enhance methane production by methylotrophic methanogenesis.

Generally, the paper is well written, albeit a few areas that need some clarification for better readability (see inline comments). The study and findings are of interest to the field as the authors point out, the role of iron bearing minerals in methane dynamics in salt marshes are not well studied. However, there are some issues that should be addressed before publication (see in-line comments below). Most notably, I think the authors need to be cautious with the sequencing data. Since the study presented sequences at the start and end of the incubations, comparisons and linkages to explain the geochemical trends (i.e., methane production) is limited to the geochemical trends at the start and end. For example, in the discussion, the authors discuss the low abundances of methylotrophic methanogenesis at the end of the incubations despite the higher methane production rates that were calculated. However, figure 3 shows that the methane production rates are low or below detection by the end of the incubations. I suggest the authors focus the discussion to the bounds of the what the dataset can provide.

We thank Anonymous Reviewer #1 for their thoughtful comments which have improved the presentation and interpretation of our manuscript. We have modified our discussion to take into account their main concern, that we are more careful to limit the connections we draw between sequence data and methane production rates over the course of the incubations because we do not have sequence data for the time in the incubations with the highest methane production rates. We include responses to line-by-line comments below with reviewer comments in italics.

**In-line comments:**

L:45: I would also cite here a recent publication: Liu, Jiarui, et al. "Iron oxides fuel anaerobic oxidation of methane in the presence of sulfate in hypersaline coastal wetland sediment." Environmental Science & Technology 59.1 (2024): 513-522.

We appreciate this suggestion of a recent paper and have added the reference.

L59: Sulfate-reducing bacteria

This has been changed.

L61-62: Salt marshes is plural but only one study is mentioned. Are there other studies to cite?

Additional references have been added to this sentence.

L129: What do the authors mean by "conservatively discarded"? I think the words "conservatively" should be removed. Which soluble oxidants are the authors concerned with here?

We see how this sentence was not sufficiently clear. The entire sentence was revised, "conservatively" has been removed and a list of potential soluble oxidants has been added.

*L130:* Can you please provide the details of the plastic bag?

Details of the bag were added.

*L150: How long did the batch incubations last?*

The number of days that the incubations lasted was added.

L152: Can you expand a bit on how the sediment was recovered by centrifugation? Why was the supernatant discarded?

We believe this confusion may be a result of including the initial steps for DNA extraction and magnetics sample preparation in the Methods section for incubations. We have moved the last three sentences (including the sentence about centrifugation) to the methods sections for DNA

extraction and magnetics measurements to make the reason for centrifugation easier to see (that preparation for magnetics measurements required freeze-dried sediments and centrifugation was done to remove most of the water first). We believe this also improves the flow of the Methods sections that were modified and appreciate the reviewer identifying this issue.

L160: Please provide details on standards for methane determinations.

We used serum bottles with matched headspace and liquid volumes for methane concentration standards to account for the partitioning of methane between gas and dissolved phases. These details have been added to the text.

L208-209: I can't quite follow this sentence, perhaps it's my lack of experience with CV but the sentence could be clearer.

The first three sentences of the paragraph were re-written to improve clarity in response to this comment and one by Reviewer #2.

L215-220: I think these few sentences are more discussion points. I would suggest moving to the discussion

We agree with the reviewer and thank them for pointing this out. Ultimately we concluded that these points were a geochemical discussion about the sediment core itself and not critical for describing or discussing the results of our incubations. We have therefore eliminated these sentences from the manuscript.

L219-220: I am a little confused by the last sentence in this section. What microcosms are the authors referring to?

See response to comment below.

L221: So the "microcosmos" are the incubation bottles? I suggest picking nomenclature and stay consistent throughout. These types of incubations you describe are also call batch incubations by some.

We see how this can be confusing and thank the reviewer for pointing this out. All instances of "microcosms" are now changed to "incubations" so that consistent language is used to describe them throughout the manuscript.

L228-229: There is something wrong with this sentence. Please clarify.

We have rewritten this sentence and revised the entire section to improve readability.

L232: I seem to be missing how the methane production rates were calculated? According to the methods methane development was only measured in the headspace of the incubation vials.

Although this isn't wrong, it doesn't account for the methane that is still in the sediment slurry. Are the rates reported here taking that portion of methane still in the slurry. This maybe small compared to the headspace but one could be underestimating the methane production rates. This should be clearly stated in the methods and also discussed.

The reviewer is correct to point out that we did not fully explain how dissolved methane was accounted for. Thank you for pointing out this oversight. We created standard bottles with the same gas-to-liquid ratio but with DI water in the liquid phase instead of slurry so that dissolved methane is accounted for in our calibration curve. This has now been explained more clearly in methods section 2.5.

L236-237: I don't see peaks at 25 hrs. I see peaks at just after 20 hrs and what I am assuming to be 27hrs.

We thank the reviewer for pointing out that our initial description had too much rounding to be accurate. The sentence has been revised to say day 28. (The figure shows days rather than hours but the numerical inaccuracy indeed needed correction.)

L240: Wouldn't it make more sense to either move the result text of the iron (II) production to the previous section (i.e., section 3.1)?

We considered moving this part of the text from Section 3.2 to 3.1 with the results of other inorganic analyses as the reviewer suggested. But we ultimately decided against it because Section 3.1 describes the geochemical results of the intact core before it was used for incubations and Section 3.2 describes the geochemical trajectory of the incubations. It makes more sense to us to keep the description of the intact core separate from the description of how methane and Fe2+ production changed over the course of the incubations than it does to put all inorganic analyses in one results section and methane concentrations in another.

L347: This paragraph has a lot of text that might be more suitable in the discussion. I would consider to move discussion like sentences to the discussion for better readability.

Given the sequential line numbering of comments and sequence of topics addressed by Reviewer #1, we assume the line number is a typo here and the reviewer means to refer to the last paragraph of Results Section 3.4 because Line 347 is in the Discussion. If so, we see the reviewer's point and have deleted the sentences that describe the metabolic capabilities of bacterial families in favor of keeping similar sentences that were already in Discussion Sections 4.1.2 and 4.2.

L339: Genomics is a little bit out of my wheelhouse; however, it is not totally clear to me what the difference is between figure 5 and 6. I understand that figure 6 looks at the archaea and bacteria separately and figure 5 looks at selected archaea and bacteria. And both look at their

connection with the added organics and minerals. Is there a reason why a cluster dendrogram is not applied to figure 6 to be more comprehensive and potentially link other groups that are impacted by the addition of the carbon and mineral substrates? For example, genus Canidatus Omnitrophus seems to be very dominant in the presence of MMA and Syntrophotalea in the presence of ethanol but these are not in Figure 5. I realize that the groups in figure 5 are at the family level and the groups in figure 6 are at the genus level so the groups I pointed out may fall in one of the families. Again, this is beyond my expertise but perhaps it would be worthwhile the authors add a sentence or two to be clear what each of these figures are trying to show, so that readers that are not experts in genomics can follow along better.

Figure 5 shows the entire microbial population at the higher taxonomic level of family to illustrate microbial diversity in the entire community in the incubations while Figure 6 separates archaea and bacteria at the taxonomic level of genus to show detail that we could not include in a more comprehensive figure like Figure 5. Since cluster dendrograms would have made Figure 6 very large, we include the dendrograms for this figure as Appendix Figure B1. We now note this explicitly in the caption for Figure 6. Depending on editorial guidance about figure size, we are willing to combine Figures 6 and B1 so that it is in a similar format to Figure 5 as suggested by this comment. We have also added a couple of explanatory sentences at the beginning of Section 3.4 to explain what is shown in Figures 5 and 6.

L368-370: I don't discount this interpretation, however, is there any evidence in the authors data to support this claim (i.e., the methane production rates)? It is kind of shocking that the authors amended the sediments with 20 mM mono-methylamine, which is considerably higher than what has been previously reported from salt marsh porewaters and yet the genetic abundances of known methylotrophic methanogens are pretty low. The methane production rates in figure 3 also show that methane production was low by the end of the incubation. Is it possible that by then the MMA was mostly consumed and thus lower methane production rates and a shift in the microbial communities to have less methanogens? Do the authors have any sequencing data from the 10 hr timepoint where methane production rates were the highest according to figure 3? It would be interesting to see what the methanogenic communities look like then compared to later in the incubation. I suggest the authors be cautious with their 16S data since it appears the data reflects the communities at the beginning of the incubation and at the end.

We sincerely wish that we could have gone back and sampled for DNA at 10 hrs in our incubations! This would have to be our biggest regret with this study and future work will aim to sample at the peak of methane production. But we do understand the point that Reviewer #1 is making. It is possible that methylotrophic methanogens could have been more abundant earlier in the incubation when methane production rates were highest. We have revised the text throughout this paragraph to be more careful about not linking the maximum rates of methane production to the snapshot of microbial diversity that we have for the end of the incubations.

L374-376: Something is off with the structure of this sentence.

The sentence has been revised to improve clarity.

L390: Are there other more recent citations that could be added here to have a more balanced reference list?

An additional citation has been added.

L395-403: I think the authors could connect more of the archaeal data here with the literature review to make the connects between ethanol degradation, acetogenesis and methanogenesis. For example, making connections between the knowledge of the relationship of Methanococcus and ethanol fermenters could be better connected with the findings in the 16S data.

We add an additional sentence that connects ethanol degradation by Syntrophotalea to the 16S rRNA abundance of Methanococcus, but not to this location. It is added in the next paragraph after the abundance of Methanococcus is discussed. This allows us to avoid repeating a sentence with the abundance of Methanococcus in both paragraphs.

L407-409: I am not sure I understand what this sentence is talking about. How did Methanosarcina succeed in the EtOH incubations?

The word "success" is replaced with "proliferation" in this sentence. The preceding sentences describe the high abundance of Methanosarcina and offer a potential explanation for this high abundance. We interpret this comment to mean that "success" can have many meanings, not just in terms of abundance, so we changed the terminology to be more specific.

L411-413: What evidence do the authors have to suggest this potential relationship between Methanosarcina and Anaerolineaceae? Is there literature that supports this claim? Why not other methanogenic groups?

This comment makes us realize we did not adequately explain our rationale for the hypothesis that we propose. We propose this potential relationship between Methanosarcina and Anaerolinacea because Methanosarcina can be an acetoclastic methanogen and such a relationship has been reported with another acetoclastic methanogen, Methanothrix. But such a relationship with Methanosarcina has not been reported in the literature yet. We revised this sentence to clarify that this potential relationship could be with another acetoclastic methanogen.

L415-429: There are a few issues in this paragraph that should be clarified here. Firstly, I do not see the presence of ANME-3 in either Figure 5 or 6, so how do we know this group was actually detected?

We have broken up this comment to address the two parts of it individually. Indeed ANME-3 were not abundant enough to be represented individually in Fig. 6 because they fell below the 5% threshold to be represented individually. This percentage threshold has been added to the text and also included in the legend of the figure. ANME-3 is included in the "Other" bar. We do include the percentage of sequences that were ANME-3 in the sentence itself. For a more detailed look at the low-abundance ASVs, the sequence data and code are publicly available through NCBI and through the hansonlabgit server.

I do not doubt that the presence of sulfide oxidizers in the 16S indicates sulfide oxidation, but without other geochemical data, at best it is only a suggestion that sulfide oxidation is active as 16S does not provide evidence of metabolic activity. Not to mention, sulfide oxidation needs electron acceptors like oxygen or nitrate, are there indications that these incubations got exposed to oxygen or was there a source of nitrate that was added? These would be crucial for sulfur cycling. Additionally, since the incubations used sulfate-free artificial seawater the sulfate in the incubations would be diluted by at least 60% because two-thirds of the slurry was sulfate-free artificial seawater which could conceivably impact the sulfate-reducers, especially in a closed system.

I suggest here the authors tone down the language of these interpretations given the evidence presented. I still get a taste of what the authors are talking about with "complex web of interspecies interactions", however, I think the evidence here is speculative without other parameters. This could be a good place to mention what future investigations should target to get a better understanding of the interactions.

We totally agree that this interpretation is speculative and it was our intention to present it that way given that we do not have geochemical measurements supporting sulfur cycling but our language turned out more confident than we thought it was. Such geochemical measurements may not have been able to resolve such cycling anyway. We now acknowledge this explicitly in the paragraph.

Our experimental setup was designed to reduce sulfur redox cycling as much as possible and we were surprised to find sulfur oxidizers and reducers represented as much as they were in our sequences. We have added more qualifiers to this paragraph to make it clear that we are offering a potential alternate explanation for the lower measured methane production rates besides less methane production. Methane may have been lost through oxidation in our bottles during the experiment, but we are not certain this occurred.

L417: Non-competitive substrates such as MMA directly fuel the cryptic methane cycle, not EtOH. If EtOH is fueling the cryptic methane cycle that is a new finding. Could the "underestimated methane production rates" be because of the calculations of the rates? The measurements were taken from the headspace but do the rates include what methane is still in

the slurry? Additionally, could the underestimated rates be because the EtOH was depleted or consumed by other organisms?

It seems to us that the meaning of the term "cryptic" in our usage, i.e. concurrent production and oxidation of methane, is different from the meaning intended in this comment. Indeed it does not have a clear definition. We changed our wording to specifically describe both methane production and consumption in our bottles rather than "cryptic" methane cycling. As described above, the methane concentrations do include the methane dissolved in the slurry. If the rates were lower because EtOH was consumed by other organisms, this would support the main explanation that there was lower methane production overall, not that there was oxidation of methane which masked the true methane production rate (the alternative explanation offered in this paragraph). We also revised some language early in the paragraph to make it more clear that lower methane production due to EtOH consumption by other organisms is more likely.

L480: Iron-reducing bacteria needs the hyphen.

Hyphen has been added.

L482-485: I find this sentence about the lack of magnetite found in the MMA incubations confusing. The title of this section suggests a discussion on the magnetite formation in the Ethanol incubation. I recommend the authors focus this section more on what's going on in the incubations with ethanol for better readability.

We have revised the sentence so that it is written more simply. We include this detail about MMA to show that reduction of ferrihydrite did not lead to magnetite formation in both cases, just with EtOH amendment. We limit discussion of MMA incubations to making this point only.

L489: It is interesting that the typical iron-reducing bacteria were not found in your analysis. Though I do see the presence of sulfate-reducing bacteria in the 16S analysis. There is literature out there that have shown sulfate-reducing bacteria can also perform iron-reduction. Have the authors considered that sulfate-reducers are implicated with iron cycling in your incubations?

This is a very good point and we thank Reviewer #1 for this observation. We have added a sentence with this possibility in this discussion.

**Reviewer #2**

The manuscript of Block et al, describes interesting incubation experiments where the effects of different iron minerals and carbon substrates on methane production and microbial communities in salt marsh sediments were studied. They identify potential connections between ferrihydrite, magnetite, hematite with C-sources, and methanogens and fermenters. The samples are derived from salt marsh cores and vertical profiles, measured over 1 cm, with mm resolution of geochemical properties was made which was impressive. The samples were incubated and the various properties measured over time. My main comment is that much more information in the methods is required on how the incubations were set up in order to interpret the results. Since no sulfate was added in the artificial seawater slurries, this may have selected against the sulfate reducing bacteria and selected for methanogens. It appears oxygenated artificial seawater (no reducing agent) was mixed with reduced anoxic sediment, in a 3:1 ratio (3x oxic seawater compared to anoxic sediment) which would have a huge effect on the natural state of the sample (microbes, porewater, and minerals would be oxidized). How does the 20 mM concentration of the added C substrates compare to the in situ concentrations? Its also unclear if there was a headspace and if it was purged. Also, the 3:1 mixture of ASW: sediment surely effected the activity because it is increasing the water content significantly, the authors should discuss more critically in the text on the potential influence of all these factors.

We thank Dr. Orsi for his constructive comments and efforts to improve this manuscript. His comments made it clear to us that the description of our incubation setup and of experimental rationale needed to be better described. These incubations did not closely mimic natural conditions. The conditions of the incubations were manipulated to investigate controls on methane production other than competition with sulfate-reducing bacteria whose activity can be dominant in salt marshes but which doesn't explain methane emissions patterns at high salinity. No sulfate was added to the incubations to intentionally minimize the activity of sulfate-reducing bacteria and select for methanogens and iron reducers. While the headspace was purged of oxygen, the liquid phase of the slurry was not and reducing agent was not added to avoid abiotic Fe(III) reduction. Oxidized iron minerals were added to all but control incubations so there was more potential redox energy to be used by the microbial community than would be true if these were simply anoxic, euxinic conditions by design. We believe the manuscript will be easier to follow and interpret after incorporation of these changes.

In retrospect, we also notice how the abstract may set up an expectation that we investigate sulfate reduction. The abstract has been significantly revised so it better represents our study.

Here are some specific comments on the text:

line 63: Are there really no sulfate reducing bacteria that can use methyl amines? Please cite a paper(s) showing this to be true. Or was this shown by the two refs at the end of the sentence?

Did they really test every single sulfate reducing bacterium ever cultivated and confirmed that none of them could use methyl amines?

We see this point and agree. The sentence is a more confident statement given that not all sulfate reducing bacteria have been tested for methylamine utilization and some may use methylated substrates. We have modified to make the statement more qualified and added a reference that describes bacterial and archaeal groups that are currently known to be important for methylamine utilization in the environment.

lines 128-140: There is some critical information missing here on how the incubations were set up. Was there a head space and if yes, was it purge of any residual O2, or was O2 left in the headspace without purging?

This was a long comment so we have separated both the comment and our responses by line breaks here to make it easier to follow. The details of the headspace in our incubations were included in the same sentence as other information about slurry composition and this may have buried the details a bit. We have revised this sentence to be three separate sentences to be easier to follow. Specific description of the headspace purging has been added as well.

lines 128-140 continued: Was a reducing agent applied to the artificial seawater to make it anoxic, since the pore waters of the core are anoxic (Fig 1)? If no reducing agent was added (none is listed), then oxygenated artificial seawater was mixed with highly reduced and sulfidic sediment. this surely had an effect on the redox potential, microbial communities, and mineral speciation. This needs to be critically discussed and acknowledged.

No reducing agent was added to the slurry to minimize the potential for abiotic reactions with our iron oxide mineral amendments that would otherwise be difficult to distinguish from mineral transformations that resulted from microbial activity. Our goal with these incubations was not to preserve in situ conditions but to manipulate them and document the microbial response. We changed redox conditions intentionally by adding iron oxide minerals but left out other oxidants like nitrate and sulfate which would also not have been present in the sediment depth interval used for our incubations. But we suspect the change in redox conditions is less dramatic than this comment predicts given that we did purge out headspace O2. Through the modifications made to address the earlier and later parts of this comment, we do more fully describe that the incubations did not start with any oxygen in the headspace and that the redox conditions were changed as part of the experimental design in order to provoke a microbial response which was then characterized with respect to methane production, iron reduction and mineral transformation, and shifts in the microbial community.

lines 128-140 continued: And, finally the justification for not adding sulfate to the artificial seawater needs to be made. Why was sulfate left out? This surely had an effect on sulfate

reducers, that depend on sulfate, and compete with methanogens for certain substrates. It needs to be discussed in the manuscript, because this is a central feature of the study.

As this comment highlights, using sulfate-free artificial seawater was a central component of our experimental design. Sulfur cycling is active and often the dominant driver of organic matter degradation in salt marshes, but the presence and concentration of sulfate does not predict methane emissions well at higher salinities. The goal of this study was to investigate other controls on methane production. We intentionally disadvantaged sulfate-reducing bacteria by starving them of sulfate to better characterize the potential of other microbial metabolisms to influence methane production by salt marsh microbial communities. Even so, we found microbial community members that were likely participating in sulfur cycling. Reading this comment made us realize that this was not explained in the manuscript. We have therefore added an additional sentence with this rationale at the first mention of sulfate-free artificial seawater in the methods section.

lines 128-140 continued: Was O2 measured in the incubations over time, to see how the original mixed slurries with the artificial seawater developed? e.g., whether the incubations ever went anoxic or not? It would be really important to know if there was residual O2 present in the incubations or not.

Because the initial headspace had no oxygen (80:20 N2/CO2) and oxygen would not have been generated by microbial activity under the conditions of these incubations, we had no reason to measure headspace oxygen. The initial slurry likely retained some dissolved oxygen that was not removed by headspace purging. But considering that, in the intact core, all oxygen in the ~2 cm of overlying water within the core barrel was consumed within the few hours between collecting the core collection and measurement by CV (described in Results Section 3.1), such a small amount of dissolved oxygen would not have persisted for very long. We now see that these initial conditions may have been unclear in the original manuscript. The description of headspace purging and filling have been expanded in response to the comment above, and hopefully this addition resolves this concern.

lines 138-139: How does the 20 mM concentration of the added C substrates compare to the in situ concentrations? Is this a realistic concentration for these substrates, that the natural communities are exposed to in nature?

One of the challenges of this sort of study is that there are few published measurements of methylamine concentrations in salt marshes (Mausz and Chen, 2019 cited in the manuscript and Fitzsimmons et al., 2023, citation below). There have been no studies of methylamines in Great Marsh, DE. The only measurements that we can find (Flax Pond in MA, USA and Grip Heath in Dorset, UK) were not reported along with salinity which makes it difficult to use these measurements to estimate methylamine concentration in Great Marsh, DE. Salinity would influence how much osmolyte was produced by marsh organisms and our site is polyhaline

(salinity up to 30 PSU) which would predict high production of osmolytes. The few reported measurements of methylamine concentrations show up to 1 mM for individual methylamines in pore water from marsh sediments. This is less than our 20 mM amendment but not by multiple orders of magnitude. We were not able to find any ethanol concentrations reported for marsh sediments but we agree our amendments are likely much higher than ethanol concentrations would be. However, we would also note that geochemical measurements of metabolite concentrations can be misleading. Some pools of chemical constituents can have a low concentration but high turnover, like acetate. An incubation without an in situ source of such a metabolite would have to start with a higher concentration that could be measured in the natural setting. We don't know enough about amine concentrations in salt marshes to tune our experiments that carefully. Instead, our choice of 20 mM of carbon substrate was made because it is the same as that used in earlier published studies investigating iron-bearing minerals and methane production (Kato et al., 2012 and Kato and Igarashi, 2018). It is also within the range of carbon substrate amendment used in the many other similar studies cited in the manuscript.

References not included in the manuscript:

Fitzsimons, M. F., Tilley, M., and C. H. L. Cree. 2023. "The Determination of Volatile Amines in Aquatic Marine Systems: A Review." Analytica Chimica Acta 1241 (340707): 340707.

Kato, S. and K. Igarashi. 2019. "Enhancement of Methanogenesis by Electric Syntrophy with Biogenic Iron-Sulfide Minerals." MicrobiologyOpen 8 (3): e00647.

lines 195-200: can you please state the figures here in the methods where those results can be found?

Figure numbers have been added to the corresponding methods sections.

General comment on the incubations: I read that the cores extended to 22 cm, but only 2-16 cm were used for the incubations. Why?

Both this comment and one from Reviewer #1 indicates to us that we did not sufficiently explain why we chose the depth interval of sediment used for our incubations. We have modified the text to explain this more clearly. The sentence describing how the top of the core was discarded is expanded and contains more detail about the rationale for not using the top 2.5 cm. We also now include explanation that the volume of sediment from 2.5 - 16.5 cm in the core was sufficient for our experiment, i.e., sediment from 16.5-22 cm was not needed. The volume of sediment required to make our slurry was calculated before core extrusion.

lines 206-208: I don't see in Figure 1A, where organic bound Fe(III) is measured.

Fe(III)-organic compounds are not individually quantified in Figure 1A. The broad shaped signals from -0.4 to -0.9 V indicate the reduction of Fe(III) to Fe(II) in Fe(III)-organic compounds and nanoparticulate material (Taillefert et al., 2000, 2002b). These signals cannot be

quantified with a standard because all Fe(III)-organic materials give a different current versus concentration curve as well as a different potential for the reduction of Fe(III) to Fe(II) (Taylor et al., 1994; Lewis et al., 1995). These signals show that Fe(III)-organic compounds and nanoparticulate are present; thus, when present, only the current measured can be reported. All peaks in the range that organic-Fe(III) aggregates and nanoparticles are found were summed in Figure 1A as Total Fe(III). There was a short description of this in the figure legend but the sentence and the figure legend has been rewritten to make this more clear.

**References not included in the manuscript:**

Taylor, S. W., G. W. Luther, III and J. H. Waite. 1994. Polarographic and spectrophotometric investigation of iron(III) complexation to 3,4-dihydroxyphenylalanine-containing peptides and proteins from Mytilus edulis. Inorganic Chemistry 33, 5819-5824. http://doi.org/10.1021/ic00103a032

Lewis, B. L., P. D. Holt, S. W. Taylor, S. W. Wilhelm, C. G. Trick, A. Butler, G. W. Luther, III. 1995. Voltammetric estimation of iron(III) thermodynamic stability constants for catecholate siderophores isolated from marine bacteria and cyanobacteria. Marine Chemistry 50, 179-188. http://doi.org/10.1016/0304-4203(95)00034-O

Figure 3: Are the multiple replicates per time point shown as individual points (and dashed vs. solid lines)? For example, there are three replicates for each treatment? Can you add some text in the legend to specify if these are biological replicates (replicate bottles)

The word "replicate" has been inserted such that it now reads "rates for each replicate incubation bottle shown..."

4a and 4b: Figure If there are replicates available please plot error bars onto the histograms for each category, and report the number of replicates (n=), and report then if the histograms displayed is an average across the replicates.

These measurements-data presented are not replicates; replicates are not standard practice in the field of environmental magnetism as error bars on the actual measurements are typically over two orders of magnitude below the measured values and too small to be plotted. Error can be introduced from humans for any of these measurements, but for 4a, the majority of variation in calculated saturation values comes from the processing of the raw data. We cite Jackson and Solheid (2010) for this processing methodology in Section 2.7. While replicates were not measured, data are robust because for all samples, multiple measurements were collected and very similar data was noted. However, the applied fields and step-sizes were distinct so they should not be averaged.

lines 284-287: Could this fine grained magnetite (what do you mean fine grained?) be biogenic magnetite produced by magneto tactic bacteria that were growing in the incubations? Biogenic magnetite in such bacteria is nano-sized.

We agree that these questions about whether the magnetite could be biogenic are important ones to address. The magnetic techniques used here are particularly sensitive to magnetite in the nanometer to micron-scale. However, there are different types of biogenic magnetite (e.g. biologically mediated versus extracellular) requiring more interpretation and literature context than could be included in a results section. The discussion of these data in Section 4.3 provides more detail and context about the grain size of the magnetite and the likelihood that it is produced by magnetotactic bacteria (which does not seem to be the case). Therefore we have decided to delete the sentence here to keep all discussion of grain size (as an interpretation based on coercivity using empirical calibrations) in Section 4.3.

Fig 5 (lines 329-331: Can you dive any deeper into your data, to search for magneto tactic bacteria that grew in the different incubations. A simple search in the OTU table for genera or species starting with "magneto" will reveal if any were detected (all magnetotactic bacteria have a genus or a species name beginning with "magneto.." There are also magneto tactic representative of the Omnitrophica as well (doi: 10.1038/s41396-018-0098-9)

Given the known magnetotactic members of the Omnitrophus genus, we were also optimistic that we would find magnetotactic bacteria in our sequence data but we were not able to identify any definitively. We cannot be sure that this Omnitrophus is magnetotactic but it might be involved with IET. We followed this suggestion and performed a search for "magnet" among the genus and species of our ASVs, but none were identified. Given that our magnetics data is not consistent with production of magnetosomes, we focused on Fe(III) reducing bacteria using FeGenie (Garber et al., 2024). Our original search did not turn up any known candidates (as described in Discussion section 4.3). We do appreciate the reference suggested by this comment and have added it to the manuscript in the discussion about potential magnetotactic bacteria in Section 4.3

Fig 5 (and lines 322-325, lines 366-367): I think that your data suggests that Bathyarchaea might be involved in methanogenesis with MMA as a methane source during methanogenesis. Even if they themselves are not making the methane (as you conclude on lines 366-367), they could be involved as syntrophic partners with other methanogens. Your results point to such an archaea-archaea syntrophy possibility.

We thank Dr. Orsi for this suggestion, an archaea-archaea syntrophy would be very interesting. We have added this potential mechanism of methane production at the end of Section 4.1.1.

lines 333-336: Acidobacteria are also a very important sulfate reducing group in salt marsh sediments (doi.org/10.1038/s41396-019-0373-4). Please report whether this group was also present in the data, and at what percentage.

Our incubation conditions intentionally selected against sulfate reduction although we did find 16S rRNA sequences for sulfur oxidizing and reducing bacteria, indicating there was likely to be some limited sulfur cycling occurring in our incubations. But Acidobacteria were not detected in our 16S rRNA sequences. Because sulfate reduction was inhibited by lack of sulfate under our experimental conditions, we do not think it makes sense to add a sentence to note the absence of Acidobacteria in the context of this study.

line 336-337: This is hard to believe, also because as I wrote above there are magneto tactic bacteria in the Omnitrophica which are found in your data. Moreover, magnetotactic bacteria are very common in these settings, as the authors state in their introduction supported by numerous previous studies. Figure 6B cannot be showing all of the data, there must be many other groups that are not shown here. The legend says "Most abundant bacteria...", please plot all of the data (including any OTUs related to genera or species starting with "Magneto...") or state what "most abundant" means and what cutoff was used to exclude low abundance groups from the figure. If there are really no "Magneto..." genera or species in the 16S, perhaps this has to do with how the incubations were set up - was O2 purged out of the headspace completely? If they were completely anoxic incubations this could select against magneto tactic bacteria that are generally micro-aerophilic. Can you please comment on this, also in the manuscript text.

We were surprised to not find any well-known iron reducers or magnetotactic bacteria too. Although we did find bacteria of the Omnitrophus genus, they were not of the known magnetotactic species (Kolinko et al., 2016). The MTB identified by Lin et al. (2018) indicate their diversity has not been well-characterized, and the reviewer's suggestion does highlight an oversight. However, there are currently 16 lineages suspected to synthesize magnetosomes and carry out magnetotaxis based on the presence of multiple magnetosome biogenesis (mam) genes in MAGs (Goswami et al., 2022). Furthermore, there is strong evidence that the magnetosome genes have been horizontally transferred multiple times as phylogenies of mam genes and standard sets of single copy genes are incongruent (Uzun et al., 2020). Given this, searching for one lineage by partial name as suggested is likely to generate both false positive and false negative inferences. As such, we have taken out this statement. We have also added the abundance cutoffs as percentages to Figure 6. These cutoffs are also included as numbers of ASVs in the Figure 6 legend.

References not included in the manuscript:

Goswami P, He K, Li J, Pan Y, Roberts AP, Lin W. 2022. Magnetotactic bacteria and magnetofossils: ecology, evolution and environmental implications. NPJ Biofilms Microbiomes 8:43.

Uzun M, Alekseeva L, Krutkina M, Koziaeva V, Grouzdev D. 2020. Unravelling the diversity of magnetotactic bacteria through analysis of open genomic databases. Sci Data 7:252.

lines 369-370: How do you know the population size is "small", I didn't see any quantitative measurements of microbial populations sizes (16S is not quantitative).

Indeed our writing was inadequately precise here, thank you for pointing this out. We have revised the sentence to be more specific about the small proportion of methanogen sequences rather than the small population of methanogens.

line 394: What do you mean "fermentative families"? Please provide a citation showing there are only fermenters, and no sulfate reducers or iron reducers in this group. Or, alternatively revise the statement to indicate you mean the group is represented by many known fermenters.

We thank Dr. Orsi for pointing out that this sentence needed to be tightened up. The sentence has been revised accordingly. There was a sentence that was originally in the Results section but removed in response to Reviewer #1's comment that it was too "discussion-y". This sentence now takes the place of the "fermentative families" phrase because it was more specific and included the appropriate references.

lines 404-410: Since you did not add sulfate in your artificial seawater slurries, this may have selected against the sulfate reducing bacteria and selected for methanogens. Please comment in the text on this.

Indeed our intention was to select against sulfate-reducing bacteria to investigate the potential for other conditions and metabolic processes to control methane production. We have added a sentence describing this to the description of incubation conditions in the Methods.

lines 460-470: There are also magneto tactic representative of the Omnitrophica as well (doi: 10.1038/s41396-018-0098-9). Some of the biogenic magnetite produced in original the salt marsh could be coming from these organisms, potentially?

Indeed this is true, some of the magnetite originally in the marsh could have been produced by a member of the Omnitrophica. There are three different magnetic components identified in the initial sediment (Fig. 4c); the mid-coercivity one has coercivity values and dispersion parameters consistent with "soft" magnetosomal magnetite, but is also within the range for detrital magnetite (Egli, 2004; Kopp and Kirschvink, 2008). Further tests would need to be performed to definitively identify if magnetosomal magnetite are present in the initial sediment but that was

beyond the scope of this study so we have not modified the text to discuss this issue in more detail.

References not in paper:

Egli, R. (2004). Characterization of individual rock magnetic components by analysis of remanence curves, 1. unmixing natural sediments. Studia Geophysica et Geodaetica, 48(2), 391–446. https://doi.org/10.1023/B:SGEG.0000020839.4530

Kopp, R. E., & Kirschvink, J. L. (2008). The identification and biogeochemical interpretation of fossil magnetotactic bacteria. *Earth-Science Reviews*, 86(1-4), 42-61. https://doi.org/10.1016/j.earscirev.2007.08.001

---

## Author Response (AR2)

We thank the associate editor, Dr. Rhew, for his review of our responses and for inviting a correction to our article. After reading the comment copied below in italics, we agree that he caught an oversight of ours. Our response is below the comment.

The response to a comment by Reviewer 1 brought to light the possibility of a slight bias in the methane production calculation. The reviewer writes: "L232: I seem to be missing how the methane production rates were calculated? According to the methods methane development was only measured in the headspace of the incubation vials. Although this isn't wrong, it doesn't account for the methane that is still in the sediment slurry. Are the rates reported here taking that portion of methane still in the slurry. This maybe small compared to the headspace but one could be underestimating the methane production rates. This should be clearly stated in the methods and also discussed." The authors respond: "The reviewer is correct to point out that we did not fully explain how dissolved methane was accounted for. Thank you for pointing out this oversight. We created standard bottles with the same gas-to-liquid ratio but with DI water in the liquid phase instead of slurry so that dissolved methane is accounted for in our calibration curve. This has now been explained more clearly in methods section 2.5."

According to the added text, the incubations and methane standards were both prepared in 125 mL serum bottles, with 75 mL headspace. The difference was in the 50 mL solution: whereas the incubations had 50 mL of a slurry containing 1:3 v/v ratio of homogenized marsh sediment to artificial sulfate-free seawater, the methane standards had 50 mL distilled water.

Methane solubility in distilled H2O may differ from that in a saline slurry. For a given temperature, the solubility of methane in distilled H2O is greater than in seawater (e.g., Yamamoto et al, 1976, Solubility of methane in distilled water and seawater, J. Chem and Eng. Data, 21, 1, p 78-80). Perhaps the solubility of methane in distilled H2O is similarly greater than in a saline sediment slurry. If so, then the incubation headspace concentrations calibrated to the distilled H2O-filled standards would yield an estimate of total methane that is too high (rather than too low if there was no liquid at all). For example, for a distilled H2O bottle, say 96% stays in the headspace and 4% goes into the solution. For the incubation, say 98% stays in the headspace and 2% go into the solution. Measuring the headspace of the incubation based on the distilled H2O calibration curve would then yield a total CH4 estimate of ~102% of actual. The actual correction factor may be trivial compared to other experimental uncertainties, but a potential bias should be acknowledged somewhere, even if briefly.

We do acknowledge the fact that distilled water will not have the same solubility as seawater or our seawater/sediment slurry, although we do expect this difference to be

small relative to other sources of variability in our experiments. We do agree it should be addressed in the text and have added the following sentence to the methods description for methane measurements:

"Although the solubility of methane in our slurries will not be the same as in distilled water, we expect these differences to be smaller than other sources of uncertainty and variability in our incubations."